# Equity and reliability of public electric vehicle charging stations in the United States

Qiao Yu [1], Tristan Que[2], Lara J. Cushing [1], Gregory Pierce[3], Ke Shen[4], Mayank Kejriwal [4], Yuan Yao [1] & Yifang Zhu [1] ✉

Equitable coverage and reliable operation of electric vehicle charging stations (EVCSs) are crucial for a just transition to a carbon-free future. Yet, a comprehensive national analysis of public EVCSs across different communities is lacking in the United States. Here, we utilize real-world reviews ($n = 470,142$) from a user-generated content platform to analyze public EVCSs at the census tract level. We find that disadvantaged communities (DACs) have 64% fewer public EVCSs per capita than non-DACs. This disparity rises to 73% when considering renters in multi-dwelling units. Additionally, EVCS users in DACs and urban areas experience significantly more reliability issues compared to those in non-DACs and rural areas, primarily related to hardware and technical failures. Given the limited access to home charging in DACs and their underserved public infrastructure, these findings highlight critical equity concerns and call for targeted investment in EVCS infrastructure and reliability improvements, particularly in DACs.

Transportation decarbonization is essential for achieving climate goals. Electric vehicles (EVs) can substantially reduce greenhouse gas emissions and contribute to improved air quality and health co-benefits[1–4]. The International Energy Agency (IEA) projects that by 2030, the global EV fleet will reach 236 million vehicles[5]. The development of a reliable and widespread electric vehicle charging station (EVCS) network is essential to meet the growing demand for EVs and achieve carbon neutrality in the transportation sector. The IEA projects that the global number of public electric vehicle supply equipment (EVSE) ports will exceed 15 million by 2030 and reach nearly 25 million by 2035. In the United States, approximately 900,000 public EVCSs will be needed by 2030 and 1.7 million by 2035[6] to support ambitious goals for EV adoption. The U.S. National Electric Vehicle Infrastructure Program[7] and the Charging and Fueling Infrastructure Grant Program[8], both stemming from the Infrastructure and Investment and Jobs Act[9] (also known as the Bipartisan Infrastructure Law) will provide $7.5 billion in funding. The Inflation Reduction Act[10] also provides considerable incentives to promote clean energy initiatives, including support for EV charging infrastructure for disadvantaged communities (DACs).

DACs in the United States are communities identified as bearing a disproportionate share of environmental, health, and climate-related burdens[11,12]. These include factors such as energy burden, pollution exposure, public health risks, and vulnerability to climate change[12]. Such factors often disproportionately affect low-income and minoritized racial and ethnic groups. DACs have a greater demand for public EVCSs, due to the higher prevalence of multi-dwelling units (MDUs) and renters within these communities. DAC residents and minoritized racial and ethnic groups are more likely to rent their home and reside in MDUs[13–15]. Although more than 80% of charging needs are predicted to be met through at-home or residential charging[16], the reliance on public charging stations varies significantly on the basis of the housing type. While only 4–16% of U.S. EV drivers with detached houses use public charging, this percentage increases dramatically to 31–81% for drivers who do not have access to private or semi-private parking facilities[17]. In addition, renters frequently lack both the financial means and property rights necessary to install private charging stations. A recent study[18] reported that over half of EV buyers will come from the low-to-middle income—households in Los Angeles County making $75,000 or less in 2019 dollars, which reflects the county's median

[1]Department of Environmental Health Sciences, Fielding School of Public Health, University of California, Los Angeles, Los Angeles, CA, USA. [2]Department of Computer Science, University of California, Los Angeles, Los Angeles, CA, USA. [3]Luskin Center for Innovation, University of California, Los Angeles, CA, USA. [4]Information Sciences Institute, University of Southern California, Marina del Rey, CA, USA. ✉e-mail: yifang@ucla.edu

household income—by 2035, many of whom do not have access to home charging. Research also indicates that EV can reduce energy and maintenance costs, particularly benefiting low-income and minoritized racial and ethnic groups who face high fuel expenses[19,20]. Without access to public charging infrastructure, the opportunity to address the transportation cost burden might be lost, further exacerbating existing disparities.

Despite the critical need for equitable charging infrastructure, equity analysis of EV charging accessibility or coverage has been limited, and even fewer studies have been conducted at the national scale. A 2019 study analyzed EV adoption and EVCS distribution in California and reported that the number of public EVCSs per 1000 households is 0.92 and 0.67 for non-DACs and DACs, respectively[21]. Another 2021 study analyzed access disparities for public EVCSs in California across different race and income groups and found that, neighborhoods with a majority of Black and Hispanic populations had the lowest likelihood of accessing public EVCSs[22]. Their likelihood of access was half that of white majority neighborhoods. A similar pattern was found for low-income neighborhoods, especially those with more MDUs. More recent studies continuously show that low-income, historically underinvested areas, and communities with higher proportions of racially and ethnically minoritized residents than white residents have significantly less access to EVCS compared to affluent, predominantly white communities. A 2022 study investigated access to public EVCSs in New York City at the zip code level[23] and reported that low-income, Black-identifying, and underserved communities had less access to public EVCSs compared to higher-income (households in New York City making $64,000 or more in 2019 dollars, which reflects the county's median household income), white, and non-DACs. In Austin, Texas, local analyses[24] found pronounced disparities, with most public chargers concentrated in non-Hispanic white neighborhoods, underscoring regional patterns of unequal access by race and income. Another study[25] tracking EVCS distribution from 2010 to 2022 found that, despite a general increase in stations, low-income populations—already underserved by public transportation—continue to lack sufficient EVCS coverage. Similarly, one national study[26] found that areas with a higher percentage of racial minorities are less likely to have access to charging stations, while more affluent regions benefit from greater access. Large-scale survey[27] data have also highlighted the role of socioeconomic factors such as income, age, and housing type in shaping EV adoption and charging behaviors, emphasizing the need for integrated policies addressing both housing and transportation to foster equitable access. A comprehensive review[28] of national EVCS distribution reinforced these findings, pointing out the persistent gaps in access for lower-income and minority communities and calling for policy frameworks that prioritize equitable infrastructure development to meet 2030 transportation electrification goals. Another recent study[29] highlights that income and racial/ethnic inequalities in charging infrastructure distribution are significantly greater than those for gas stations, with disparities varying widely across states and urbanized areas. Overall, the gradually expanding body of research on EVCS equity has consistently revealed a striking fact: inequities in EVCS access persist, even as the number of stations continues to increase. This underscores the need for further research on equitable EVCS distribution.

Studies mentioned above often sourced public EVCSs data from the U.S. Department of Energy's (DOE) Alternative Fuels Data Center (AFDC)[30]. The U.S. DOE defines EVCS as a location that houses one or more EVSEs, such as a parking garage or a mall parking lot. An EVSE, which is sometimes referred to as an EV charging port, is designed to deliver power to a single vehicle at a time. AFDC collects public EVCS data, including attributes such as location, accessibility, hours of operation, and pricing, directly from various charging point operators

(CPOs), each of whom may use different reporting criteria. As a result, some EVCSs lack complete information, and certain stations classified as 'public' may, in fact, be workplace charging locations that are not accessible to the general public. Owing to the potential misclassification of EVCSs with access restrictions, the actual gap in public charging stations could be more substantial than reported. The true potential for increased EV use in DACs also depends on whether residents can freely access these public EVCSs without additional parking fees, highlighting the need for more validated real-world usage data to accurately assess accessibility. To address this, the California Energy Commission has enhanced its public EVCS location data by integrating input from a user-generated content platform in California[31]. However, nationwide integration of this type of data is not available, as the U.S. DOE does not incorporate user-generated content into their public EVCS datasets.

Moreover, charging reliability significantly influences the user experience and is a critical factor in the adoption and sustained use of EVs. A dissatisfying charging experience can even deter current EV users, potentially causing them to revert to gasoline vehicles[32]. Despite its importance, EVCS operation data are often treated as proprietary business secrets, which restricts access for researchers. This limitation poses a challenge in evaluating the reliability, operation, and maintenance status of charging stations across the nation. Despite public concerns[33] and industry reports[34] on the reliability of public EVCS, there is a notable gap in research focusing on charging station reliability and user experiences, especially in DACs.

This study aims to address the gap in our understanding of national-level public EVCS coverage and reliability assessment, examining the distribution of EVCSs and analyzing user experiences across various communities, with a special emphasis on DACs. We begin by using an EVCS data platform with user reviews to analyze distributive equity in EVCSs across all census tracts in the United States. We then employ large language models (LLMs) to conduct a sentiment analysis of real-world user review data from the platform on public EVCS experience. We subsequently apply grounded theory to conduct qualitative analysis and identify issues encountered by EVCS users. The results of these analyses demonstrate that those who have limited at-home charging opportunities have less reliable public charging infrastructure coverage, highlighting significant equity issues.

## Results
### Public EVCSs in the United States
As shown in Fig. 1a, the total numbers of EVCSs from both the Department of Energy (DOE) and a popular online user-generated content (UGC) platform are similar. The DOE database contains a total of 59,826 EVCS data points, with 93% (55,792) of which are labeled as public.

With more detailed user input and information gathered from the UGC platform, we filtered out approximately 2% (1106) of the stations that were under repair, 27% (16,290) of the stations that had access restrictions (such as employees only or hotel patrons only), and 10% (6202) of the stations that required additional parking fees. This filtering process left 61% (37,536) of the stations that were fully public, without any restrictions or additional payments needed, ensuring that the UGC data accurately reflect EVCS that are genuinely accessible to the public.

We further analyzed the public EVSE data from the DOE (93% of the records) and the UGC platform (61% of the records) as shown in Fig. 1b. The UGC data show a similar distribution of Level 1 (up to 1.9 kW power), Level 2 (2.9 to 19.2 kW power), and direct current (DC) fast chargers (25–350 kW power) as compared to the DOE data, while providing more detailed site information and user reviews. We found that currently, most of the EVSEs are Level 2 (78–80%), with 18–21% having DC fast charging capability.

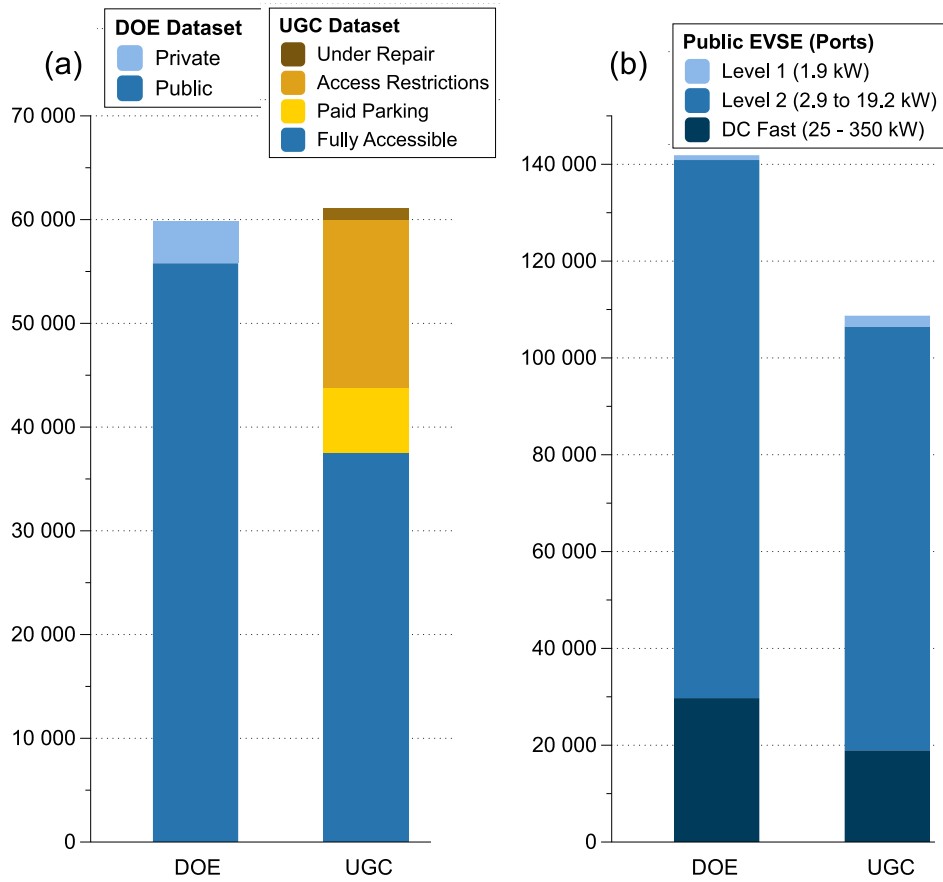

**Fig. 1 | Data comparison between the Department of Energy (DOE) database and the user-generated content (UGC) platform. a** Electric vehicle charging stations (EVCSs), and (**b**) public electric vehicle supply equipment (EVSEs).

## EVCS coverage disparity

We define equitable EVCS coverage as a 3-mile radius around charging stations that are open to the public and have no additional parking fees, based on data from the UGC platform. Studies indicate that most public EV charging events occur within this distance for both DC fast chargers and Level 2 chargers[35]. This radius also aligns with the preferred driving distance for EV drivers seeking a charge[36,37]. Using data from a UGC platform, we mapped EVCS locations through a kernel density heat map to account for edge effects, recognizing that EV drivers may cross census tract boundaries to access charging stations.

Our findings reveal significant disparities in per 1000 population EVCS coverage between DAC and non-DAC, as shown in Fig. 2a. Out of the 48 states and the District of Columbia, 24 states showed no statistically significant difference between DAC and non-DAC areas, largely due to the limited number of EVCSs available for analysis. In the remaining 25 states with significant results, 19 exhibited disparities, with DAC areas having markedly lower public EVCS coverage. Overall, DACs in the contiguous United States have, on average, 64% fewer public EVCSs within the 3-mile radius coverage.

This disparity is even more significant when considering the needs of residents in MDUs and renters, who typically have less access to private charging options and thus a greater reliance on public EVCSs. In these contexts, 28 states out of the 29 states with significant results show disparity. DACs have 73% fewer EVCS coverage per capita when adjusted for MDU and renter populations (Fig. 2b).

We also found that, compared to simply counting EVCS numbers in each census tract, our spatial coverage approach revealed greater disparities. As shown in Supplementary Fig. 1, when using EVCS numbers, the disparities are 26% per 1000 population, and 60% per 100 renters living in MDU. After adjusting census tract size, we found that

nearly all states exhibit a higher density of public EVCSs per square mile in DAC areas (Supplementary Fig. 2). This pattern likely arises because DACs are often situated in dense urban environments with smaller census tracts, resulting in a higher EVCS density per square mile. Yet, as we showed earlier, per capita disparities persist, and EVCS coverage remains insufficient to meet the potential needs of DAC populations. Supplementary Table 1 provides additional detail on state-level disparities across all metrics, including adjustments for tract size and population (e.g., population density). The table offers further context on how EVCS coverage varies by state, with notable disparities persisting in many DAC areas even when accounting for population density.

## EVCS reliability

In addition to inequitable public EVCS spatial coverage, we also found statistically significant differences in mean sentiment scores (e.g., whether the experience with an EVCS is positive) between DAC and non-DAC, as well as between urban and rural areas, as shown in Fig. 3a. The average sentiment score, ranked in percentiles among all communities, is significantly lower in the DAC than in the non-DAC. In other words, users in the DAC are more likely to have a negative experience with public EVCSs. Similar to another study[38], users in urban areas report lower sentiment rankings compared to those in rural areas.

Using grounded theory (see Methods), we analyzed a comprehensive set of negative EVCS user reviews (those expressing dissatisfaction or reporting issues) to identify and categorize the main issues (see Table 1) for all communities. These categories were then used to generate prompts for an LLM, which classified all the negative reviews ($n = 128{,}271$). Figure 3b illustrates the distribution of public EVCS issues, broken down by category percentage. Among the most

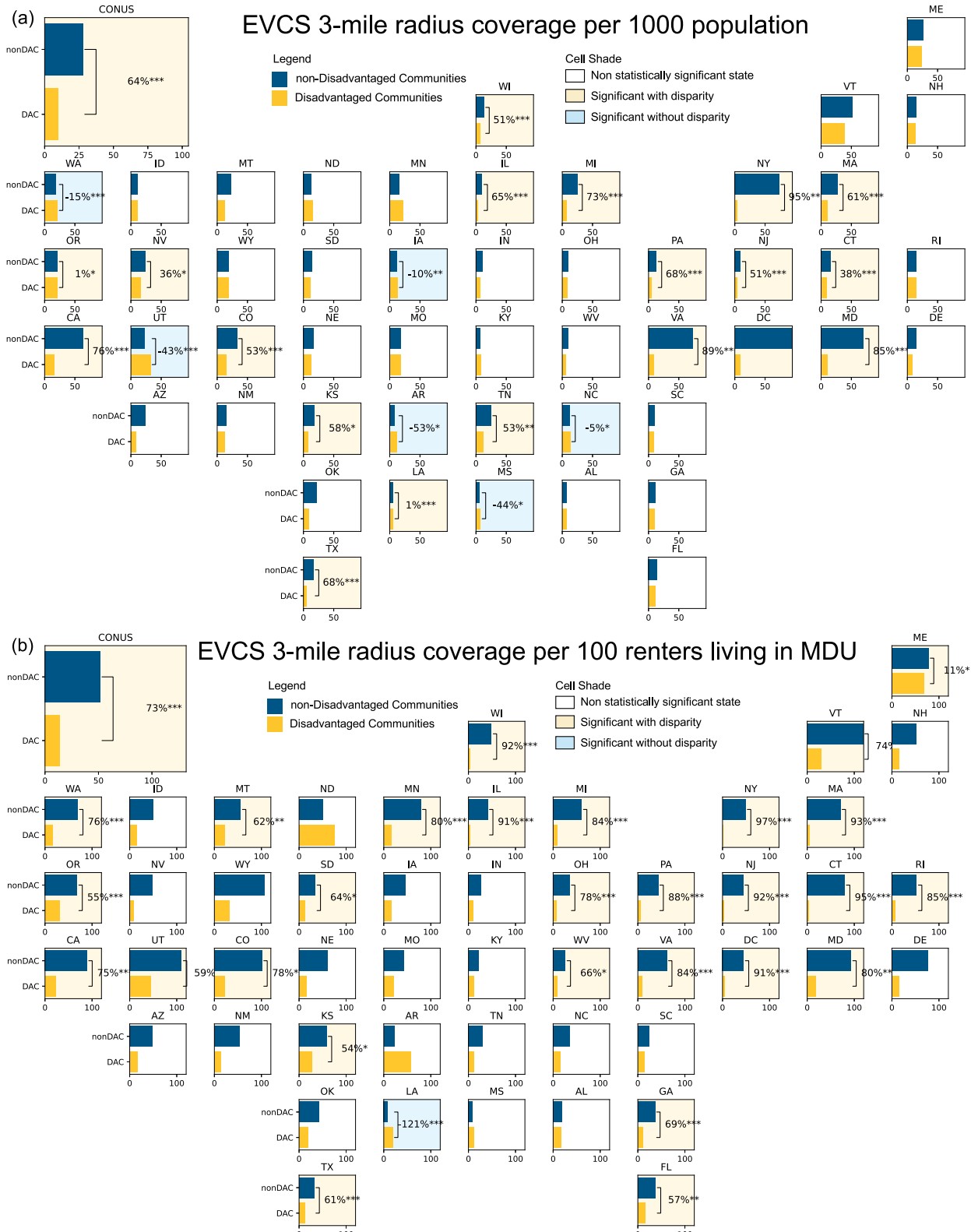

**Fig. 2 | Mean public electric vehicle charging station (EVCS) 3-mile radius coverage in 2022, comparing disadvantaged and non-disadvantaged communities. a** Coverage per 1000 population, and (**b**) coverage per 100 renters living in multi-dwelling units. The yellow columns represent disadvantaged communities, while blue columns represent non-disadvantaged communities. The light yellow state cells indicate states with statistically significant disparities, the light blue state cells indicate significant but without disparity, and the white cells represent non-statistically significant states. Disparity percentage is calculated by dividing the difference between disadvantaged and non-disadvantaged communities by the value in non-disadvantaged communities. Statistical significance was evaluated using the Mann–Whitney $U$ test (two-sided). ***$p < 0.001$, **$p < 0.01$, *$p < 0.05$.

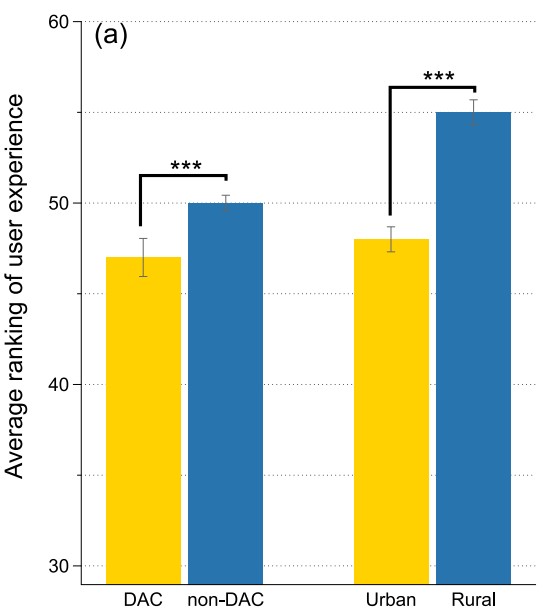
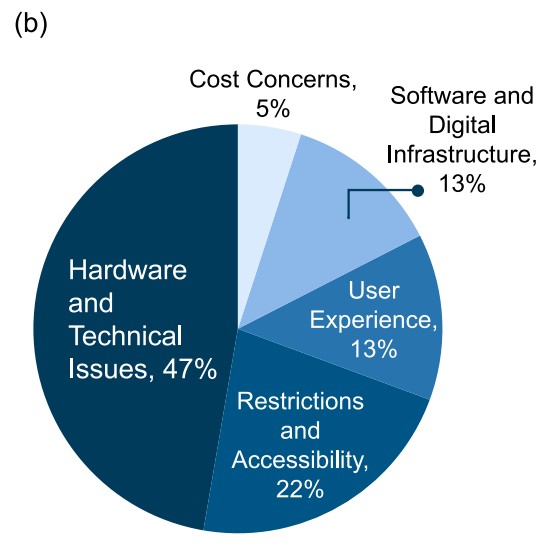

**Fig. 3 | Users experience rankings and problem categorization results for public EV charging stations. a** Non-disadvantaged communities (non-DAC) and rural areas exhibit significantly higher average rankings of positive user experience, as compared to disadvantaged communities (DAC) and urban areas (***$p < 0.001$; exact $p$-values: $2.7 \times 10^{-7}$ for the comparison between DAC and non-DAC areas, and 0.00051 for the comparison between urban and rural areas). User experience is ranked by sentiment scores, which are derived from each user review. Statistical significance was evaluated using the $T$-test (two-sided). The error bars indicate 95% confidence intervals. **b** Distribution of different categories of problems encountered by public EVCS users.

**Table 1 | Summary of current issues for electric vehicle charging stations (EVCSs) based on negative user reviews**

| Category | Issue | Description |
|---|---|---|
| 1. Hardware and technical issues | Charger functionality | Malfunctions with chargers, screens, card readers, and plugs. |
| | Adapter compatibility | Issues related to the physical connection between the EV and the charging station, including the availability and compatibility of adapters. |
| | General failures | Broken parts, power failure, and physical damage impacting the charging process. |
| | Interrupted charging sessions | Charging stops unexpectedly. |
| 2. Software and digital infrastructure | App Functionality | Technical problems within apps, such as crashes and bugs affecting user interaction and charging station operation, or App showing incorrect charging availability information. |
| | Connectivity problems | Challenges with internet connection at charging stations, affecting functionality and access to services. |
| 3. User experience | Slow charging | Issues related to lower-than-expected charging speeds. |
| | Capacity | Long wait times for charging or requests for more charging stations |
| | Customer support | Availability and quality of support for troubleshooting, guidance, and resolving charging issues. |
| | Physical infrastructure and amenity gap | Discomfort from unsheltered setups in inclement weather, inadequate lighting, or the lack of nearby amenities like restrooms and food services. |
| | Network dependency | Challenges due to the need for network-specific apps or payment cards for charging access, restricting ease of use. |
| 4. Restrictions and accessibility | Parking and access limitations | Customer-only or employee-only restrictions, no entry, parking permit requirements, time restrictions, high-cost paid parking and difficulty locating chargers. |
| | Blocked charging spots | Issues with ICE vehicles or EVs not charging occupying spots. |
| | Limited parking for different EVs | Parking spaces that are too small for vans and larger EVs, limiting accessibility for diverse EV users. |
| | Signage and policy enforcement | The need for clear signage indicating spots are designated exclusively for EV charging, guide signs to direct users to EV chargers, and enforcement measures to prevent ICE (Internal Combustion Engine) vehicles from blocking these spots. |
| 5. Cost concerns | High pricing | Perceived as too expensive, especially in comparison to previous free services or when compared to the cost of gasoline. |
| | Pricing transparency | Confusions raised from discrepancies between displayed and billed charging costs. |

prominent issues, hardware and technical issues were reported by 47% of the users, who frequently mentioned malfunctions with chargers, screens, card readers, and plugs, as well as issues related to adapter compatibility and general failures impacting the charging process.

Software and digital infrastructure problems were also substantial, as noted by 13% of the users, particularly issues with app functionality, such as crashes and incorrect charging availability information, along with connectivity problems at charging stations that further

exacerbated user frustration. In terms of user experience, 13% of the users cited common issues, including slow charging speeds, long wait times, and the availability and quality of customer support. The users also noted discomfort from unsheltered setups, inadequate lighting, and a lack of nearby amenities. Restrictions and accessibility posed major challenges for 22% of the users, with reports of parking and access limitations, blocked charging spots by internal combustion vehicles or noncharging EVs, and limited parking for larger EVs. This category also highlighted the need for clear signage and policy enforcement. Finally, cost concerns were raised by 5% of the users, who perceived charging services as too expensive and expressed confusion over pricing transparency. These findings underscore the multifaceted challenges that EVCSs face, highlighting areas for improvement to enhance the overall user experience and support the broader adoption of EVs.

## Discussion

We present a national analysis of public EVCS coverage from an equity perspective, using user-generated data that more accurately reflect public accessibility and charger reliability. We applied LLMs to streamline the sentiment analysis of a substantial volume ($n = 470,142$) of review data. With input from real-world EVCS users, we have found that despite several policies targeting the development of public EVCS in DACs, significant disparities still exist nationwide. Previous studies[21,23,39] relying solely on DOE data were either unable to detect these disparities or identified them only at the local and state levels.

The issue of charging infrastructure in the DAC is complex and multifaceted. At-home charging, often considered the predominant solution for EV charging, presents many challenges for DAC residents. This is particularly problematic as a study[40] found that more than 46% of households in DACs may require updates to their electrical service panels for higher amperage to accommodate at-home EV charging and other electrification needs, but such upgrades often are not prioritized. Additionally, while single-family homes (a standalone residential structure, both detached and attached such as townhouse) typically have less demand for public charging due to their potential to support overnight charging[41], the prevalence of accessory dwelling units (ADUs) in the DAC—often implemented to support affordable housing—makes at-home charging complicated. It is common for multiple families (either unrelated households or extended family members from different generations living as separate units) in DACs to share a single-family home, often utilizing ADUs with the same property[42,43]. At-home charging may be insufficient in such high-density living arrangements, increasing the need for accessible public charging infrastructure. Despite having greater needs for public charging than non-DAC residents, significant disparities in public EVCS coverage persist across nearly all DACs with sufficient sample sizes for analysis as illustrated in Fig. 2.

One viable solution to mitigate the limited availability of overnight charging in DACs is the strategic deployment of more DC fast chargers. Currently, as illustrated in Fig. 1, these chargers constitute only 18% of the total charging infrastructure. Increasing the proportion of DC fast chargers could enhance charging accessibility and efficiency, compensating for the lack of private charging options in these areas. Yet, implementing DC fast chargers will require a reliable power infrastructure. Residents of DACs experience more frequent and longer power outages than others do, especially during extreme weather events[44–46]. Such frequent outages could hinder DAC residents' willingness to adopt EVs or clean energy in general[47]. Prolonged power outages can impede the effective charging of EVs, which are essential for mobility and accessing vital services. The increased frequency and duration of power outages in these areas underscore the vulnerability of existing infrastructure, which is often outdated and underfunded for necessary upgrades and maintenance. This perpetuates inequality, leaving these communities less resilient against both routine and extreme weather events that are increasingly common due to climate change[48,49]. To address these issues, it is essential to prioritize infrastructure improvements in DACs, ensuring that they receive the necessary upgrades to withstand power disruptions. Additionally, integrating renewable energy sources and storage solutions could provide these communities with more stable and independent power supplies, reducing their dependency on the traditional grid and increasing their resilience to outages[50–52].

Reliability, or the performance of public EVCS, is another crucial aspect of infrastructure effectiveness. Instead of sending out surveys to collect user reviews on EVCS experience, we explored how LLMs can substantially streamline the analysis of the public EVCS user experience by directly analyzing existing online reviews. The traditional approach to sentiment analysis on user reviews can be labor-intensive and costly; for example, manually labeling sentiment in 3000 user reviews takes approximately 25 h, and problem categorization tasks, being more complex, require even more time. In contrast, using LLMs to process the same batch of 3000 sentiments takes only a few minutes and costs between $0.50 and $2.5, depending on the chosen LLMs and the length of the reviews. LLMs have demonstrated utility across various fields such as education[53], medicine[54], and clinical research[55]. This study shows their effectiveness in analyzing secondary qualitative data to assess public EVCS reliability. By leveraging the LLMs, we find that the public EVCS user experience is significantly lower in the DAC and urban areas.

Our sentiment analysis aligns with a previous study[38] regarding differences in user experiences with urban and rural public EVCSs, indicating that urban areas often report lower user satisfaction. One reason for this could be that individuals in rural settings, such as those near camping sites or national parks, typically have lower expectations. For example, users commented: "Very pleased to see an EV charging station here in the park. Thanks for making this possible!!!" or " Loved discovering a charging station during our camping trip! Will come back for more adventures!" This difference in expectations could partially explain why urban users report more negative experiences, as they frequently face access restrictions, longer charging times, and hardware issues. Similarly, since DAC areas are often located in densely populated urban areas, this could also explain why public EVCS in these regions tend to have lower sentiment scores. However, this does not negate the need for improvements in charging infrastructure in rural areas. A recent study[56] has shown that nearly 60-80% of census tracts across the United States have no public charging access, except for residents living near highway charging corridors. The average travel time to EVCS in these regions are longer than in urban areas. Even with corridor charging infrastructure programs, rural areas still remain underserved[57].

Our research identified that hardware and technical issues are the most frequently mentioned problems encountered by users of public EVCSs. This finding highlights the urgent need to increase maintenance funding, as exemplified in small part by programs such as the recent Electric Vehicle Charger Reliability and Accessibility Accelerator[58]. This program allocates $100 million specifically for the repair and upkeep of public EVCSs, addressing a critical gap in the current infrastructure and ensuring that these essential services remain functional and accessible. Additionally, access restrictions present substantial barriers, as many stations are not available to the general public. Funding stipulations should enforce public accessibility. Another critical aspect is the network dependency of charging stations. Both the National Electric Vehicle Infrastructure[7] and California Electric Vehicle Infrastructure Project[59] have recognized this by requiring CPOs do not limit user access to specific charging network subscriptions, which is a step in the right direction. Charging speed has also emerged as a key concern, particularly in DACs where residents often spend more time commuting and working across various locations[60–62]. The existing infrastructure may not adequately

support the rapid charging necessary for efficient use in these communities.

While cost concerns may not seem paramount, it is important to recognize that early adopters of EVs are typically wealthier individuals[16,21,63]. Households without access to residential charging face higher overall costs of EV ownership, as non-residential charging options tend to be more expensive[64]. Research indicates that EVs can address future transportation needs and reduce energy and maintenance costs, particularly benefiting low-income and minoritized racial and ethnic groups who face high fuel expenses[19,20]. Yet, higher charging prices might exacerbate existing energy insecurity, especially for Black and Hispanic households[65]. Thus, policies such as providing subsidized charging programs to DAC residents could provide a great opportunity to address historical burdens and advance equity in these communities. This approach could further lower barriers to EV adoption and ensure that all community members can benefit from the transition to EVs.

A recent study[66] demonstrated a strong correlation between the availability of EVCSs and the adoption of EVs in DACs. The study revealed that increasing the number of public charging stations in DACs could have a nearly threefold greater impact on EV adoption than non-DAC areas. This finding aligns with our analysis of the physical characteristics of housing, which indicates that individuals in non-DAC areas have less reliance on public charging due to the availability of private residential charging options. Another study[67] reported that charging $100 per month for a designated overnight parking space with charging capabilities could reduce people's willingness to adopt EVs by approximately 65%, highlighting the dilemma faced by DAC residents who often lack access to designated parking spaces needed for overnight home charging. Our study, along with previous literature, underscores the critical need for public EVCSs in DACs and the importance of addressing these disparities.

In the discussion concerning infrastructure development, a well-known chicken-and-egg dilemma frequently arises: the notion that without adequate infrastructure, there is little incentive to adopt EVs, yet without substantial EV adoption, the expansion of infrastructure may seem unjustified. Challenging this perspective, particularly in the context of DACs, we argue the importance of prioritizing the establishment of publicly accessible EVCSs—the "chicken"—to stimulate EV adoption—the "eggs." By emphasizing infrastructure development initially, we can prevent DAC residents from being marginalized in the clean energy transition, thereby addressing and mitigating existing disparities. By initially emphasizing infrastructure development and coupling it with equitable incentive policies for EVs[68–71], we can make EVCSs more accessible, reliable, easy to use, and affordable. This comprehensive approach will prevent DAC residents from being marginalized in the clean energy transition and pave the way for a cleaner energy future.

## Limitations

In our analysis, we focused primarily on spatial coverage, while considering whether EVCS locations are publicly accessible and free of additional parking fees. However, we recognize that "access" to EVCS could be defined more comprehensively, incorporating factors such as charging speed, station capacity, and user behavior (e.g., average charging time and frequency of use). Access to user behavioral and historical charging session data could enhance understanding, though this information is often proprietary and difficult to obtain. Future research could develop a more comprehensive analysis by utilizing historical session data to examine changes in reliability across different communities. By considering operational capacity, charger historical records, and infrastructure locations, such research could enable a more holistic assessment of EVCS accessibility.

Another limitation of this study is the unavailability of nationwide data on the number of registered EVs within each census tract. This data gap restricts our ability to assess EVCS accessibility relative to EV ownership in each community. Future studies could improve precision and apply additional control metrics that adjust for EV ownership density across tracts.

Finally, from a broader equity and social justice perspective, it is important to acknowledge that EVs alone cannot address all transportation-related challenges, despite their significant potential for decarbonizing the transportation sector in combating climate change. Research[72] has highlighted the need for comprehensive solutions, including systematic changes to reduce car dependency and develop more equitable mobility options. Additionally, concerns[73] have been raised regarding the environmental and social impacts of battery production, global waste inequality throughout the lifecycle stages, and broader justice implications of the clean energy transition—including how renewable energy development[74] affects local communities and the impacts on traditional carbon-intensive industries and their workforces[75].

Returning to this paper's focus on EV charging infrastructure, future research and policy directions must emphasize procedural justice to achieve genuine equity in infrastructure deployment. Study[76] indicates that without a comprehensive justice framework encompassing distributive and recognition justice components, the transition to EVs may perpetuate or even exacerbate existing inequalities. This is especially critical as residents of DACs are often excluded from the policy-making process during the clean energy transition[68,77]. A holistic approach[78,79] is essential, incorporating elements of recognition and procedural justice.

## Methods

### EV charging station location and review data

We gathered publicly available data on EVCS from the DOE Alternative Fuel Data Center's Alternative Fueling Station Locator[30]. We obtained data for EVCS by selecting "electric" as the fuel option in October 2022. At the time of collection, the DOE data were sourced from EVCS network providers or CPOs through their APIs. The data included information on whether the stations were public or private, as provided directly by the network providers rather than the DOE.

To obtain more comprehensive and real-world information regarding station usage, user experience, and restriction information, we acquired EV charging station location and equipment data, as well as user review data, from a popular user-generated content (UGC) platform. The data were also obtained in October 2022. This platform is widely used by EV owners to share their experiences, providing valuable insights into the practical challenges and satisfaction levels associated with various charging stations. Unlike social media platforms like Twitter (rebranded as X), which may contain bot-generated or incomprehensible comments, the EVCS UGC dataset we selected is of very high quality, as the EVCS community tends to provide meaningful input, as indicated by other research[38]. Empty comments or those containing only single letters have been filtered out, accounting for less than 5% of the data. A limitation is that, while this UGC platform effectively reaches actual EV and EVCS users—often difficult to access through traditional mail-in surveys—it may still reflect biases common to online platforms, such as demographic skew related to race, age, and income. Unfortunately, due to privacy restrictions, we cannot access detailed demographic data to adjust for these potential biases. Future work could address this limitation by integrating demographic data through anonymized data-sharing agreements if such data are available.

This research complies with all relevant ethical regulations set by the National Institutes of Health[80]. It involves the use of publicly available data posted by EVCS users and does not constitute human subjects research, as it relies on secondary data and does not include any personally identifiable information that could enable the re-identification of contributors.

We validated the total number of EVCS locations and EVSE data between the DOE database and the UGC platform and found them to be consistent with each other (Fig. 1). This validation confirmed the reliability of our data sources. All subsequent analyses are based on the detailed user experience and feedback data obtained from the UGC platform, which provides a real-world perspective on EV charging station usage and issues.

## Environmental Justice data

We obtained environmental justice and DAC designation data from the DOE Justice40 initiative via the DOE Disadvantaged Community Reporter (DCR)[81]. This dataset includes a set of indicators to identify and characterize DACs. Specifically, the DOE dataset encompasses multiple environmental, economic, and social indicators that reflect the cumulative burdens experienced by different census tracts. Examples of these indicators include air quality, energy burden, and socioeconomic factors[11].

It is important to note that the White House Council on Environmental Quality also provides the Climate and Economic Justice Screening Tool (CEJST)[82], a new tool that aims to help federal agencies identify DACs. The CEJST offers a binary classification to identify DACs, supported by detailed data attributes that indicate the type and extent of disadvantage across a wide range of indicators. The final DAC classification also includes census tracts that, while not meeting every threshold, are completely surrounded by DACs and have low-income rates at or above the 50th percentile. In contrast, the DOE classification is based on top 20% state percentile rankings derived from a low-income indicator and various categorical thresholds with a focus in energy-related indicators. We chose to use data from the DOE DCR, as it includes additional energy-related categories that may be relevant to EVCS, which could allow us to better correlate these challenges with user experiences and sentiments regarding EVCS. The substantial overlap between CEJST and DOE DAC data ensures consistency in identifying DACs.

In addition to the EJ data, we also gathered additional data from the U.S. Census American Community Survey (ACS)[83] 2010 to match the ACS version year with the DOE data. We obtained information on whether a census tract is categorized as "Urbanized Areas" to determine whether a census tract is urban or rural via the Census Geocoder[84]. Furthermore, we downloaded data on available vehicles, renters, and housing characteristics from the 2010 ACS Table B08013 and subject Table S2504. This supplementary dataset provides a broader context for our analysis, allowing us to account for urbanization levels and housing characteristics in our analysis of EVCS coverage for different populations in need.

## Geospatial data processing

To analyze the spatial distribution of public EVCS and minimize edge effects, we employed a Kernel Density Estimation (KDE) method, as used in similar studies[85,86]. KDE was performed in QGIS v3.36.0, allowing us to convert point data representing EVCS locations into a continuous density surface. This technique enabled us to identify high- and low-density areas in EVCS distribution while addressing potential edge effects that may arise from boundaries in spatial data.

For the KDE analysis, we selected a 3-mile radius for the kernel function. This radius was chosen based on previous studies[35–37], which suggest that 3 miles is a reasonable distance for locating a public EVCS for charging. By applying this radius, we aimed to capture EVCS accessibility within a practical range for daily use, especially for DAC residents without at-home charging capacities. Once the KDE density map was generated, we overlaid it with the DOE DCR census tract shapefile. Using this overlay, we applied zonal statistics to calculate the sum of EVCS density within each census tract. This sum represents the cumulative EVCS density captured in each tract, providing a quantitative measure of EVCS coverage for both DAC and non-DAC areas.

To assess the robustness of our findings, we conducted a sensitivity analysis by varying the KDE radius to 5 miles and 10 miles. As shown in Supplementary Table 2, these alternative radius values did not alter our conclusions, demonstrating that our results are consistent across different spatial scales.

## LLM sentiment analysis

To determine the sentiment of online reviews, previous studies[38,87,88] typically use natural language processing (NLP) techniques, employing models such as convolutional neural networks (CNN). The process usually involves several steps: data preprocessing, feature extraction, model training, and evaluation. On the other hand, LLMs leverage transformer architectures, which include both encoder and decoder mechanisms. The encoder processes the input text to create a rich, context-aware representation, whereas the decoder generates human-like text or performs classification tasks based on this representation. These models can reduce the expensive and time-consuming processes associated with traditional NLP methods because they are pre-trained on vast amounts of text data and can be fine-tuned to address specific tasks.

To validate the performance of the LLM model on EVCS review sentiment analysis, we first randomly selected a subset of user reviews from the entire dataset. Unlike traditional methods, which require extensive time and a large dataset to train a CNN model, LLMs can achieve high performance with pre-trained models. Previous studies have suggested that validating sentiment analysis results requires a sample size of at least 100, ideally 300, with studies ranging from 250 to 3000 reviews, representing 0.13% to 1.46% of the total dataset[89,90]. We randomly selected a sample size of 3000 reviews, which provides a balance between robustness and practical constraints.

Figure 4 shows a schematic of the data curation and model validation process. Two research assistants, acting as human experts, annotated the sentiment of review samples as either positive or negative. Similar to a previous study on EVCS sentiment analysis[38], we trained our researchers to understand the typical terminologies used in the EV charging user community and reached a consensus on what constitutes a positive or negative sentiment. The Cohen's kappa coefficient for interrater reliability was 0.81, indicating substantial agreement between the two researchers. This high interrater reliability demonstrates the robustness of our annotation process.

Next, we created a prompt for ChatGPT. Prompt engineering is crucial as it defines the scope and context for the LLM, helping it leverage its vast training data more effectively to increase accuracy[91,92]. We also set the "temperature", a parameter that controls the variability of responses generated by the LLMs, to 0 to ensure consistency in the output[93]. We used the following prompt:

"You are an assistant analyzing the sentiment of EVCS reviews. For each review provided, determine its overall sentiment and return a number indicating this sentiment: 1 for positive and 0 for negative. Please keep your response short, formatted as a single digit integer number."

Supplementary Table 3 shows the model performance on sentiment analysis using different models. The CNN model on same EVCS review dataset sentiment classification, as reported[38] by Asensio et al., achieved an accuracy of 84.7% with a precision, recall, and F1 score of 0.86. In comparison, for the LLMs used in this study, GPT-3.5 achieved an accuracy of 86.2% with a precision of 0.95, a recall of 0.81, and an F1 score of 0.88; and GPT-4 further improved the performance, achieving an accuracy of 87.1%, with a precision of 0.96, recall of 0.82, and an F1 score of 0.88. Considering that the interrater reliability (measured by Cohen's kappa in Supplementary Table 4) is 0.81, the prediction results from the LLM underscore the effectiveness of LLMs in handling sentiment analysis in EVCS reviews, providing a more efficient and accurate alternative to traditional methods. We found that, while ChatGPT-4 performed slightly better, the GPT-3.5 model—

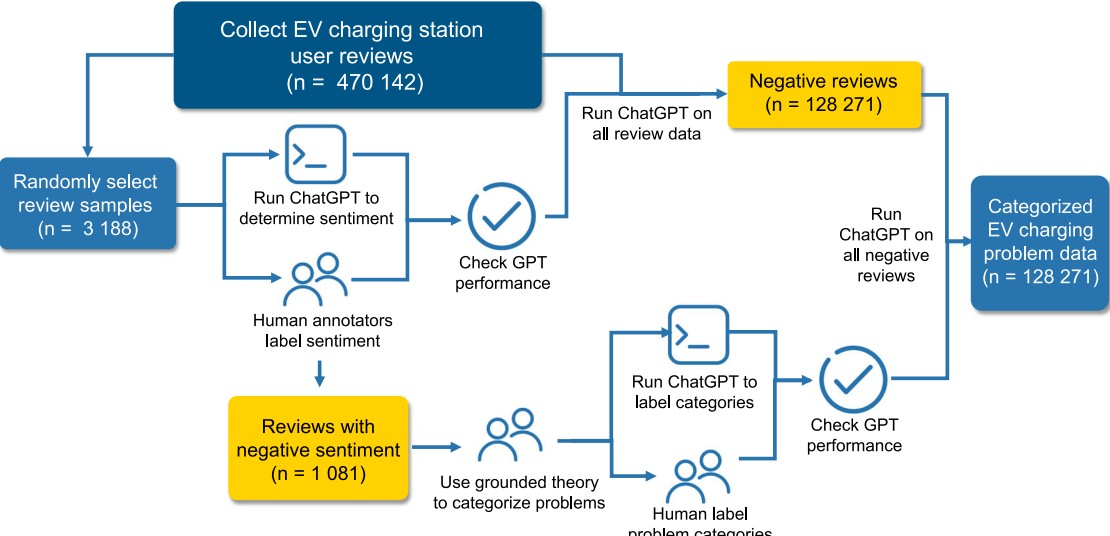

**Fig. 4 | Workflow for data curation, LLM validation, and EVCS problem categorization.** We collect over 470,000 EV charging station user reviews and apply a large language model (LLM) to determine sentiment across all reviews. A random subset of 3188 reviews is annotated by humans to validate sentiment classification. Among these, 1081 reviews are identified as negative and further categorized using grounded theory and human-labeled data to identify common problem types. LLM is then used to categorize all 128,271 negative reviews, and model performance is assessed against human-labeled categories.

20 times more cost-effective–also demonstrated strong capabilities for determining the sentiment of EVCS reviews. This highlights the effectiveness of LLMs in sentiment analysis tasks. Given that GPT-4's improvements are focused more on multimodal and visual recognition abilities, we conclude that GPT-3.5 is a more cost-effective choice for sentiment analysis in our study. After validating the model on a subset, we applied GPT-3.5 to the entire dataset ($n = 470{,}142$) to identify negative reviews.

After determining the sentiment of each review, we calculated an aggregate sentiment score for each census tract, accounting for the varying number of reviews per station. Stations with few reviews provide an insufficient data foundation for accurate sentiment analysis. For example, a station with only two reviews, both positive, would appear perfect but offers inadequate evidence for reliable comparison. To ensure statistical validity, we employed the lower bound of the Wilson score confidence interval for a Bernoulli parameter, a method frequently used in similar research contexts[94,95]. The Wilson score confidence interval is calculated using the Eq. (1):

$$Wilson\ score = \hat{p} + \frac{z^2}{2n} \pm z \sqrt{\frac{\hat{p}(1-\hat{p}) + \frac{z^2}{4n}}{n}} \Big/ \left(1 + \frac{z^2}{n}\right) \qquad (1)$$

where $\hat{p}$ is the observed proportion of positive reviews, $n$ is the total number of reviews, and $z$ is the $z$-value corresponding to the desired confidence level.

We applied a 95% confidence level to generate a more reliable sentiment score for each census tract with EV charging stations and reviews. These scores were then ranked and converted into percentile rankings on a national scale to facilitate subsequent analysis, as illustrated in Fig. 3a.

### Problem categorization of negative reviews

We used Charmaz's constructivist grounded theory[96,97] approach, a systematic methodology in social science research that involves constructing theories through interpretive analysis, to analyze the negative sentiment in the reviews ($n = 1081$) and categorize them into distinct themes. In the initial phase, two researchers independently reviewed the dataset to perform open coding. Each review was scrutinized line-by-line to identify key phrases and concepts, with the primary goal of breaking down the data into discrete parts and generating

initial codes that represent the various issues and experiences described in the reviews. This process resulted in a diverse set of initial codes reflecting the wide range of user feedback. Some example quotes and initial codes: one user stated, "The charger stopped working after a few minutes." Annotator 1 coded this as Interrupted Charging Sessions, while Annotator 2 labeled it as Charging Stops Unexpectedly. Another quote, "Had trouble connecting my car to the charger. Adapter didn't fit," was coded as Adapter Compatibility by Annotator 1 and Adapter Issues by Annotator 2. A third user noted, "The app kept crashing, and it took forever to find an available station." Annotator 1 assigned the code App Crashes, and Annotator 2 used App Functionality. Finally, one user remarked, "There was no shelter or lighting at the station." Annotator 1 labeled this as Lack of Shelter, while Annotator 2 coded it as Inadequate Lighting.

Following the initial coding, the researchers created subcategories from the generated codes. Each subcategory represented a more specific aspect of the broader themes identified during open coding. This step involved grouping related codes together to form meaningful subcategories that encapsulate distinct dimensions of the user feedback. Some example subcategories: Hardware and Technical Issues included codes such as Charger Functionality (e.g., "The charger stopped working after a few minutes"), Adapter Compatibility (e.g., "The adapter didn't fit properly"), General Failures, and Interrupted Charging Sessions (e.g., "The charging stopped unexpectedly"). Software and Digital Infrastructure encompassed issues like App Functionality (e.g., "The app kept crashing") and Connectivity Problems. The User Experience category covered themes such as Slow Charging, Capacity, Customer Support, and Physical Infrastructure and Amenity Gap (e.g., "No shelter or lighting").

After independently generating subcategories, the two researchers engaged in a series of collaborative discussions to refine and consolidate the subcategories. This involved critically evaluating each subcategory and discussing overlaps, redundancies, and unique aspects. Through this iterative process, we added, deleted, and merged subcategories as necessary to ensure that the resulting categories accurately reflected the data and were comprehensive and distinct. Some codes were merged for clarity and consistency. For instance, "App Crashes" and "App Problems" were combined into "App Functionality", while "Charging Stops Unexpectedly" and "Interrupted Charging Sessions" were merged under a single code. Additionally,

new subcategories were added to better capture recurring themes, including Network Dependency and Signage and Policy Enforcement.

The final phase of our analysis was axial coding and selective coding. This process involved exploring connections, patterns, and hierarchies among the subcategories to understand how they interact and influence each other. The goal was to integrate the review data in a way that provided a deeper understanding of the underlying issues and themes within the EVCS reviews. Six core categories were coded that represented the most integrative themes. These core categories captured the overarching patterns that structure challenges with public EVCSs.

To ensure rigor in our grounded theory analysis, we applied Quality Assurance and Quality Control practices aligned with methodology proposed by Charmaz and Thornberg[98]. We maintained a reflective approach, regularly revisiting our methods and findings to ensure they accurately represented user experiences. Initially, two researchers independently coded the reviews, followed by collaborative discussion to refine subcategories. For example, "App Crashes" and "App Problems" were merged into "App Functionality", while similar steps helped consolidate other overlapping codes. Throughout the analysis, we followed an iterative process of open, axial, and selective coding to continually reassess and refine our categories. Cross-checking between researchers in each phase helped identify necessary additions, deletions, or merges, ensuring that our categories accurately represented user feedback while avoiding redundancy. As part of our QA/QC process, we validated our findings by comparing our coding categories with three recent studies[99–101] on EVCS reliability. This step aligns with Charmaz and Thornberg's suggestion to engage with existing literature. As a result of this QA/QC process, our results aligned well with all the issues identified in prior studies and revealed additional problem categories users face in public EVCS.

### LLM for problem category labeling

After defining the problem categories, we employed a validation method similar to our sentiment analysis to assess the model's performance in problem categorization. We first reached agreements on the problem labeling criteria, such as selecting the major problem or the first problem mentioned if multiple issues were described in a negative review. Subsequently, we independently labeled the reviews and validated the LLM's performance in categorizing these problems. Given that multiple classifications can reduce inter-rater consistency compared with binary classifications, our inter-rater reliability ($\kappa = 0.71$) decreased but still remained within an acceptable range for reliability. The LLM exhibited commendable accuracy, with values ranging from 0.77 to 0.82, indicating robust performance despite the complexity of the task (Supplemental Tables 5 and 6).

We used the following prompt to run GPT-3.5 model on all negative reviews ($n = 128,271$):

"You are an assistant analyzing the problem mentioned in EVCS reviews. For each review provided, determine the problem category and return only a number indicating this. If multiple problems detected in the review, choose the major problem or the first problem. Please keep your response short, formatted as a single digit integer number".

Table 1 was also provided to the LLM along with the prompt. The full prompt and code are available in the Code availability section.

### Reporting summary

Further information on research design is available in the Nature Portfolio Reporting Summary linked to this article.

## Data availability

The EVCS location data from the U.S. DOE are publicly available from the AFDC website (https://afdc.energy.gov/stations#/find/nearest).

Environmental justice data were downloaded from DOE Disadvantaged Communities Reporter (https://energyjustice.egs.anl.gov). However, due to government changes, these data are no longer available as of 2025. A backup of the dataset has been uploaded to the Figshare repository at: https://figshare.com/s/1cca6ebb81aae425e5ba. Additional Census data are available from the United States Census Bureau (https://data.census.gov). Data from the user-generated platform contain identifiable information and cannot be posted publicly due to privacy restrictions. A de-identified version is available from the corresponding author upon request. Source data are provided with this paper.

## Code availability

All codes and prompts for sentiment analysis and problem categorization have been deposited in the Figshare repository at: https://figshare.com/s/209509bbc5067c936ce3.

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

## Acknowledgements

This work was supported by University of California Office of the President Climate Action under grant number R02CP6948 to Y.Z. The views, opinions, findings, and conclusions or recommendations expressed in this paper are strictly those of the authors. They do not necessarily reflect the views of funding agencies and/or authors' affiliated institutes.

## Author contributions

Q.Y. and Y.Z. designed the research. T.Q. performed data cleaning. Q.Y. conducted the equity and LLM analysis. K.S. and M.K. provided technical support for the LLM analysis. Y.Y. assisted with statistical analysis. Y.Z. supervised the research. Q.Y. wrote the manuscript and Q.Y., L.J. Q., G.P., and Y.Z. reviewed the manuscript.

## Competing interests

The authors declare no competing interests.
