## [Peer review file · Nature Communications]

Equity and reliability of public electric vehicle charging stations in the United States

Corresponding Author: Professor Yifang Zhu

Version 0:

Reviewer comments:

Reviewer #1

(Remarks to the Author)

This manuscript covers the topic of EVSE reliability at a national scale, which to my knowledge of the literature has yet to be considered. This topic is important and notable for many researchers working within the domains of transportation, infrastructure planning, and energy, and the breadth of the analysis presented is appropriate for publication in Nature Communications.

I do have some minor questions and comments that I believe should be addressed so that this research can be of maximum utility to those of us who work within this field. I order my thoughts sequentially using page and line numbers.

Introduction —

Page 2, lines 61-73: The authors are not incorrect that there has been a limited pool of research has been conducted at a national scale in the United States but there have been some notable articles to come out in recent years. Examples include Ermagun and Tian (2024), Carlton and Sultana (2024), and Juhasz and Hochmair (2023) — some reference should be made to these. Additionally, the individual case studies cited by the authors such as Canepa (2019) and Hsu and Fingerman (2021) are indeed major works in this area, but the authors should cite newer case studies on EVSE and equity of which there are quite a few. EV adoption and EVSE standards have changed extensively since 2019 and 2021, and the authors should note the progression of the literature.

Page 2, lines 75-100: This is a fair statement about the AFDC, however the database does contain records on hours of availability and pricing for chargers among other reliability attributes. This may be worth mentioning.

Results —

Page 3, lines 116-136: Excellent descriptives and comparisons between data sources. This should be quite helpful for replicability.

Page 4, lines 142-154: A weakness of this analysis is that it only considers the DAC status of the census tract itself, and not the neighborhood of tracts around the EVCS. If a charging station is at the very edge of a non-DAC tract, surrounded by DAC tracts, it would be categorized as non-DAC even though it is likely serving DAC communities. This weakness should be later addressed in the discussion section.

My overall impressions of the results are quite positive, and the presentation is very successful.

Discussion —

This is a very well written discussion with many thoughtful connections between findings and past studies.

Methods —

Page 13, lines 411-416: What is the author's rationale for not including the name of the "popular UGC platform" that the data was mined from? I am presuming it is PlugShare, but this should be stated as it casts doubt over the entire results. It also could lead someone to believe that the reason is it not being shared is perhaps because the API or data has restrictions on it that the authors have not mentioned.

Page 13, lines 433-443: The author's statement that the CEJST provides only binary classifications "without offering a detailed score" is not necessarily true. While CEJST is a binary measure, the CEJST data layer itself contains detailed attributes including on the type and quantity of disadvantage based on individual categorical thresholds (on topics such as housing, energy burden, and pollution) that were exceeded. Therefore, it is perfectly possible to discern the degree of disadvantage an area has by analyzing the individual scores contained within the GIS attribute table. The DAC may also be appropriate to use, but the authors should be more detailed in their representation of the CEJST.

Page 14, LLM methodology: This a very novel and interesting application of a LLM. My kudos to the authors.

Page 15, lines 507-514: Has this approach with ChatGPT been documented in other studies? If so, this should be cited. From the supplemental table, it would appear that GPT 3.5 and GPT 4 had almost the same overall accuracy and precision. Is there a statistically significant difference? This would be helpful to understand since other researchers may have more constrained resources, and if there is no statistically significant difference, GPT 3.5 would be a more parsimonious choice.

Pages 16-18, Grounded Theory: I can appreciate how detailed the authors were in documenting their process. Since Grounded Theory has different ontological and epistemological underpinnings, it would be helpful for the authors to document which version of grounded theory they are using. It would appear they are working off of Charmaz (2006), but this has not been explicitly stated. Once this has been established, the authors should mention their QA/QC methodology. A good reference for this is Charmaz and Thornberg (2021).

(Remarks on code availability)

Reviewer #2

(Remarks to the Author)

The paper offers an interesting comparison of DAC and non-DAC areas by analyzing the number of EV charging stations in census tracts and explores user experience using user-generated data. However, while the study utilized valuable datasets, it has significant shortcomings that detract from its contribution to the field and make it unsuitable for publication in its current form.

First, while the study aims to investigate "equity of access," neither "equity" nor "access" is adequately defined or thoroughly analyzed. Equity can take on different meanings depending on the perspective considered, yet the paper limits its exploration to a basic comparison of the number of chargers in DAC and non-DAC areas. This approach overlooks the complexity of equity and fails to capture its multi-dimensional nature. Furthermore, the absence of key control variables, such as population density, proximity to points of interest, the number of registered EVs within each tract, and many others, hampers the study to provide a realistic presentation of equity.

In my opinion, simply having an equal number of chargers in DAC and non-DAC areas does not necessarily reflect equitable access. Equity involves more than equal distribution and it should account for the specific needs and characteristics of each area. However, if the study considers equal distribution as a measure of equity, it must clearly define what is meant by "equity" and justify why this definition is appropriate in the context of EV charging infrastructure. Without a clear, contextually relevant definition, assuming that an equal number of chargers equates to equitable access is overly simplistic and could be misleading about the true nature of equity in this context.

The study reduces "access" to the mere number of EV charging stations without considering essential spatial factors like distance. Since census tracts vary widely in size, simply counting the number of chargers in each tract offers a limited picture. For example, in many previous studies, access is more comprehensively measured by factors like the distance to chargers, which is crucial, especially when considering the significant size variations among census tracts. The findings in this study are likely influenced by population density and the size of these tracts. It is possible that DACs are denser than non-DACs, but without controlling for tract size and density, we cannot know. This is extremely important because dense census tracts tend to be significantly smaller in size, which limits the available public space for EV charging infrastructure (e.g., parking spaces).

Furthermore, dense census tracts often have significantly short distances to neighboring tracts, which could allow residents to rely on chargers outside their tract. This further questions the analysis, as simply counting chargers within a single tract does not provide a reliable measure of access. In short, the current approach does not capture the complexity of equity in access, especially without accounting for spatial (e.g., distance to chargers) and density-related factors.

The study mentions that "we present the first national analysis of equity in access to public EV charging ...," which is not accurate. There is already a substantial body of research that has addressed "equity" and "access" to EVCS with great depth (at local and national levels). This is one other major issue of the paper, which is the lack of engagement with the relevant literature, which leaves its contributions unclear and its contextualization within the field weak. I'd suggest that the paper provide more detailed literature on studies that investigated equity and accessibility. This will also help authors come up with a clearer definition of these two terms.

In terms of methodology, the paper would benefit from a better presentation of the data sources and analytical process. The study does not provide sufficient details about the user-generated data platform, which is a critical component of the research. Additionally, the analysis would be more compelling if visualized through maps, and the inclusion of a table summarizing the datasets would benefit the method and data section.

In conclusion, while the paper has potential, especially with its use of user-generated data and LLM, it falls short in several key areas. A more robust analysis, a more comprehensive literature review and definitions of key terms, and better visualization of results are needed to meet the standards of journal publication.

(Remarks on code availability)

The provided code includes Python scripts for sentiment analysis and categorization. However, the scripts for the other parts of the analysis are not included.

Reviewer #3

(Remarks to the Author)

This paper explores issues of charging equity in the U.S. It conducts a state-wide analysis and examines the needs of disadvantaged communities, who are noted to predominantly lack adequate access to public EV charging stations. I found the article engaging until I reached the section on the use of a LLM piece. There are several arguments made that appear to have methodological flaws. The reviewer is uncertain whether this can truly be termed 'sentiment analysis,' as it feels more like binary encoding with the LLM assigning labels based on those numbers.

The authors argue for the novelty of using UGC platform and point out potential shortcomings of the NREL database. While the reviewer acknowledges the importance of including UGC, there are likely biases that should be considered, such as gender, income, age, and language due to the types of users contributing reviews. Typically, in NLP analyses, such as previous studies using Twitter data, irrelevant reviews and outliers are cleansed. This step seems to be missing in the current analysis. Because of the uncertainty surrounding the UGC dataset, it feels like the authors are attempting to combine two papers into one. The first paper should focus on addressing the use of UGC, while the second should concentrate on conducting sentiment analysis using UGC. This creates confusion for readers, as the article mixes both quantitative and qualitative research approaches.

The authors also argue that the large volume of data justifies the use of an LLM, but the reviewer disagrees with this claim. It is still possible to perform conventional sentiment analysis using NLP techniques. If the use of an LLM is to be justified, it should be compared with conventional methods not just by citing other's work that is trained on different data and research framework.

In conclusion, I am rejecting this article due to the foundational issues with its hypothesis and methods, but I would encourage the authors to further develop their work.

My additional comments are provided below:

- 1) In the introduction section, the authors should provide a clearer definition of DAC and explain how current federal policies like Justice40 address or define them. The problem statement is mainly based on empirical studies in metropolitan regions such as California and NYC. There are other case studies widely available across the U.S., and further descriptions would strengthen the problem statement.
- 2) What are the characteristics of users in the UGC platform, including factors like race, age, and income? The reason for this question is that digital platforms are typically more used by younger populations and likely English speakers. While the authors argue that the DOE database has shortcomings, doesn't UGC also carry inherent biases? How do the authors address these issues?
- 3) I liked Figure 2.
- 4) This leads to further doubts about whether the authors objectively cleansed outliers. The authors might want to refer to conventional processes for sentiment analysis in NLP. Additionally, EV drivers are mobile, and it's challenging to confirm that a user reviewing a public EV charger in California actually resides there. How did the authors account for this type of bias?
- 5) The reviewer believes that the review categories used to prompt the LLM in Table 2 should be supported not only by data but also by a comprehensive literature review.
- 6) Instead of explicitly assigning binary labels, the authors could benefit from assigning continuous sentiment scores to measure relativity on a continuous scale.
- 7) The authors mention that two reviewers were assigned to the code. Since this involves qualitative aspects, did the authors obtain IRB approval? Shouldn't the positionality of the reviewers also be acknowledged? Is it typical to have only two reviewers?
- 8) The authors selected a sample size of 3,000, but the reviewer disagrees that this is a robust sampling approach. Instead, sampling should involve multiple iterations, and the results should be generalized based on the compilation of those iterations.
- 9) Sentiment analysis results for the entire state should be presented in maps. Figure 3a currently illustrates the status of the U.S. rather than providing a breakdown across individual states.
- 10) In Table S2, the authors attempt to justify the use of GPT-3.5 or GPT-4 over conventional NLP methods and cite Asenio et al. (2020). This is incorrect. It is not appropriate to claim one model is superior to another unless they are trained on the same datasets. For example, in the article, GPT-4 achieves 87.1% accuracy, but if another study using CNN achieves 87.2% accuracy, would that mean CNN is superior to GPT-4 in sentiment analysis? The authors should conduct a separate NLP

analysis and compare it with GPT-3.5 and GPT-4.

11) This article is submitted to Nature Communications. Given the aim and scope of the journal, how can the findings be generalized to a global audience?

(Remarks on code availability)

N/A

Version 1:

Reviewer comments:

Reviewer #1

(Remarks to the Author)

I appreciate the time and effort the authors invested in addressing my comments and concerns during the review process. Their detailed revisions, including the incorporation of new references, expansion on the methodology, and careful attention to ensuring the reliability of the data and its presentation, are commendable. While I cannot speak for the other reviewers, it seems that the authors also gave due consideration to critiques raised by others, including Reviewer 3's detailed comments about the LLM. Given these thorough efforts and improvements, I am of the opinion that this article is now suitable for publication in Nature Communications.

(Remarks on code availability)

Reviewer #2

(Remarks to the Author)

(Remarks on code availability)

Reviewer #3

(Remarks to the Author)

I appreciate the authors' efforts and the time they dedicated to improving the overall quality of their manuscript. While there are some points on which I disagree, the manuscript appears suitable for publication in its current form.

I acknowledge that leveraging LLMs and AI agents is a trending topic in empirical research; however, I encourage the authors to carefully reflect on its theoretical contributions in their future work. Overall, great work.

(Remarks on code availability)

I have reviewed them.

Reviewer #4

(Remarks to the Author)

I was not one of the original referees, but the editor has asked me to review the new submission along with the reviewer reports to arbitrate between them. This is easier than I expected, but (a) the study is a first-rate contribution to the fields of energy and transport equity and (b) the authors have done a good-faith job responding to all of the earlier comments, even those from the critical referees. For these reasons, I believe the paper is of publishable quality and I recommend that the critical review be overruled.

In reviewing the manuscript, I did have a series of Minor Revisions to suggest which would improve it before it is published.

In the Abstract, the authors highlight that "disadvantaged communities (DACs) have 64% fewer public EVCSs per capita than non-DACs" and that "users in DACs and 12 urban areas experience significantly more reliability issues compared to

those in non-13 DACs and rural areas." This begs the question for me, however, which is what % of EVCS's actually experience reliability issues (especially negative user experiences) overall vs. in DACs. I say this because it's a very novel finding and also one I have experienced myself - I do not own a car but have tried renting an EV twice and experienced many EVCS reliability issues where I live in New England.

Second, the article's two core themes are equity and reliability, but these are (obviously) concerned with vehicle use and charging infrastructure. Other equity and justice considerations are manifold and include other dimensions (procedural justice, dependence on automobility, recognitional justice, even issues of mineral extraction) well beyond the vehicle or charging station. It would be worth mentioning this in the Limitations section, that the study takes a narrower focus on equity, and that future work could explore (with large datasets such as the study's) these other justice concerns. Especially the impacts from energy transition metal mining, supply chain and labor issues in EV manufacturing, and pollution flows after end of life. To support these claims, see:

Henderson, Jason. "EVs are not the answer: a mobility justice critique of electric vehicle transitions." *Annals of the American Association of Geographers* 110, no. 6 (2020): 1993-2010.

Skeete, Jean-Paul, Peter Wells, Xue Dong, Oliver Heidrich, and Gavin Harper. "Beyond the Event horizon: Battery waste, recycling, and sustainability in the United Kingdom electric vehicle transition." *Energy Research & Social Science* 69 (2020): 101581.

Sovacool, Benjamin K., Johannes Kester, Lance Noel, and Gerardo Zarazua de Rubens. "Energy injustice and Nordic electric mobility: Inequality, elitism, and externalities in the electrification of vehicle-to-grid (V2G) transport." *Ecological economics* 157 (2019): 205-217.

Sovacool, B.K., Hook, A., Martiskainen, M. and Baker, L., 2019. The whole systems energy injustice of four European low-carbon transitions. *Global Environmental Change*, 58, p.101958.

Svobodova, Kamila, John R. Owen, Deanna Kemp, Vítězslav Moudrý, et al. "Decarbonization, population disruption and resource inventories in the global energy transition." *Nature communications* 13, no. 1 (2022): 7674.

Third, the study is very up to date on its citations but it is missing one more very recent one from DY Lee and his team: Lee, D. Y., Wilson, A., McDermott, M. H.,... & Ward, J. (2025). Does electric mobility display racial or income disparities? Quantifying inequality in the distribution of electric vehicle adoption and charging infrastructure in the United States. *Applied Energy*, 378, 124795.

Fourth, in Table 1 "Summary of current issues for electric vehicle charging stations (EVCSs) 284 based on negative user reviews," it would be good to add a final column indicating how frequent these issues come up from the data (either as a total N= or as a %).

Otherwise, a solid study that I look forward to citing in my own work.

(Remarks on code availability)

Version 2:

Reviewer comments:

Reviewer #3

(Remarks to the Author)

The manuscript is suitable for publication in its current form.

(Remarks on code availability)

N/A

Reviewer #4

(Remarks to the Author)

The authors have addressed all of the concerns I raised earlier, and I now recommend the study for publication. I also look forward to citing this important paper!

(Remarks on code availability)

All comments from the reviewers have been addressed below point-by-point.

Responses are in blue font and manuscript changes are also included in this file and highlighted in yellow in the revised manuscript.

Reviewers' comments:

Reviewer #1 (Remarks to the Author):

This manuscript covers the topic of EVSE reliability at a national scale, which to my knowledge of the literature has yet to be considered. This topic is important and notable for many researchers working within the domains of transportation, infrastructure planning, and energy, and the breadth of the analysis presented is appropriate for publication in Nature Communications.

Thank you so much for taking the time to review our paper and for your recognition of its contributions!

I do have some minor questions and comments that I believe should be addressed so that this research can be of maximum utility to those of us who work within this field. I order my thoughts sequentially using page and line numbers.

Introduction —

Page 2, lines 61-73: The authors are not incorrect that there has been a limited pool of research has been conducted at a national scale in the United States but there have been some notable articles to come out in recent years. Examples include Ermagun and Tian (2024), Carlton and Sultana (2024), and Juhasz and Hochmair (2023) — some reference should be made to these. Additionally, the individual case studies cited by the authors such as Canepa (2019) and Hsu and Fingerman (2021) are indeed major works in this area, but the authors should cite newer case studies on EVSE and equity of which there are quite a few. EV adoption and EVSE standards have changed extensively since 2019 and 2021, and the authors should note the progression of the literature.

Thank you for your detailed suggestions. We have added the recommended references from Ermagun and Tian (2024) and Juhasz and Hochmair (2023) and discussed the progression of the literature in the Introduction section.

In addition, we have searched the literature and identified three recent studies (Varghese et al., Jiao et al., Lee et al.) published in 2024 and added those to the revised manuscript as well:

“In Austin, Texas, local analyses¹ found pronounced disparities, with most public chargers concentrated in Non-Hispanic White neighborhoods, underscoring regional patterns of unequal access by race and income. Another study² tracking EVCS distribution from 2010 to 2022 found that, despite a general increase in stations, low-income populations—already underserved by public transportation—continue to lack sufficient EVCS coverage. Similarly, one national study³ found that areas with a higher percentage of racial minorities are less likely to have access to charging stations, while more affluent regions benefit from greater access. Large-scale survey⁴ data have also highlighted the role of socioeconomic factors such as income, age, and housing type in shaping EV adoption and charging behaviors, emphasizing the need for integrated policies addressing both housing and transportation to foster equitable access. A comprehensive review⁵ of national EVCS distribution reinforced these findings, pointing out the persistent gaps in access for lower-income and minority communities and calling for policy frameworks that prioritize equitable infrastructure development to meet 2030 transportation electrification goals. Overall, the gradually expanding body of research on EVCS equity has consistently revealed a striking fact: inequities in EVCS access persist, even as the number of stations continues to increase. This underscores the need for further research on equitable EVCS distribution.”

References:

1. Jiao, J., Choi, S. J. & Nguyen, C. Toward an equitable transportation electrification plan: Measuring public electric vehicle charging station access disparities in Austin, Texas. *PLoS One* **19**, e0309302 (2024).
2. Juhasz, L. & Hochmair, H. H. Spatial and Temporal Analysis of Location and Usage of Public Electric Vehicle Charging Infrastructure in the United States. *ISBN Volume 11*, 83–100 (2023).
3. Ermagun, A. & Tian, J. Charging into inequality: A national study of social, economic, and environment correlates of electric vehicle charging stations. *Energy Res. Soc. Sci.* **115**, 103622 (2024).
4. Lee, D. Y., McDermott, M. H., Sovacool, B. K. & Isaac, R. Toward just and equitable mobility: Socioeconomic and perceptual barriers for electric vehicles and charging infrastructure in the United States. *Energy Clim. Chang.* **5**, 100146 (2024).
5. Varghese, A. M., Menon, N. & Ermagun, A. Equitable distribution of electric vehicle charging infrastructure: A systematic review. *Renew. Sustain. Energy Rev.* **206**, 114825 (2024).

We further expanded discussion on Carlton and Sultana (2024) in Discussion section, which was included in the original manuscript. The updates are as follows:

“A recent study¹ has shown that nearly 60-80% of census tracts across the United States have no public charging access, except for residents living near highway charging corridors. Many rural areas still lack public charging stations, and the average travel time to EVCS in these regions are longer than in urban areas.”

Ref 1: Carlton, G. J. & Sultana, S. Electric vehicle charging equity and accessibility: A comprehensive United States policy analysis. *Transp. Res. Part D Transp. Environ.* **129**, 104123 (2024).

Page 2, lines 75-100: This is a fair statement about the AFDC, however the database does contain records on hours of availability and pricing for chargers among other reliability attributes. This may be worth mentioning.

Thank you for your suggestion. We have revised our manuscript as follows:

“AFDC collects public EVCS data, including attributes such as location, accessibility, hours of operation, and pricing, directly from various charging point operators (CPOs), each of whom may use different reporting criteria. As a result, some EVCSs lack complete information, and certain stations classified as 'public' may, in fact, be workplace charging locations that are not accessible to the general public.”

Results —

Page 3, lines 116-136: Excellent descriptives and comparisons between data sources. This should be quite helpful for replicability.

Thank you! We are glad you found the descriptives and data source comparisons helpful.

Page 4, lines 142-154: A weakness of this analysis is that it only considers the DAC status of the census tract itself, and not the neighborhood of tracts around the EVCS. If a charging station is at the very edge of a non-DAC tract, surrounded by DAC tracts, it would be categorized as non-DAC even though it is likely serving DAC communities. This weakness should be later addressed in the discussion section.

We acknowledge this as a limitation, which was also pointed out by Reviewer #2. Initially, we used kernel density estimation (KDE) to address edge effects, as you suggested. However, we were concerned that converting KDE density to per capita values (e.g., EV charging stations per 1 000 residents versus EV charging coverage per 1 000 residents) might be less intuitive for the broad audience of *Nature Communications*. Nevertheless, we have now included this KDE analysis and created new Fig 2 in the revised manuscript.

My overall impressions of the results are quite positive, and the presentation is very successful.

Thank you for your positive feedback.

Discussion —

This is a very well written discussion with many thoughtful connections between findings and past studies.

Thank you for your positive feedback.

Methods —

Page 13, lines 411-416: What is the author's rationale for not including the name of the "popular UGC platform" that the data was mined from? I am presuming it is PlugShare, but this should be stated as it casts doubt over the entire results. It also could lead someone to believe that the reason is it not being shared is perhaps because the API or data has restrictions on it that the authors have not mentioned.

You are correct, the data indeed were sourced from PlugShare. We did not include the platform name in the manuscript because another research group, Asensio et al., used PlugShare data in their 2020 publication in Nature Sustainability without naming it either. Although they did not explicitly mention the platform, the data structure indicates its use. We felt it might be more appropriate not to specify the name to avoid implying an endorsement of PlugShare. At the time of our study, only a commercial license for API usage was available, and an academic license was not despite our attempts. Thus, we utilized publicly available PlugShare data using web crawlers instead of their API, which complies with the platform's guidelines, as their robots.txt file does not restrict such use for non-commercial purposes.

Reference:

Asensio, O. I. *et al.* Real-time data from mobile platforms to evaluate sustainable transportation infrastructure. *Nat. Sustain.* 2020 36 3, 463–471 (2020).

Page 13, lines 433-443: The author's statement that the CEJST provides only binary classifications "without offering a detailed score" is not necessarily true. While CEJST is a binary measure, the CEJST data layer itself contains detailed attributes including on the type and quantity of disadvantage based on individual categorical thresholds (on topics such as housing, energy burden, and pollution) that were exceeded. Therefore, it is perfectly possible to discern the degree of disadvantage an area has by analyzing the individual scores contained within the GIS attribute table. The DAC

may also be appropriate to use, but the authors should be more detailed in their representation of the CEJST.

Thank you for your suggestion. We agree that our original wording may have caused confusion, and we have removed the phrase 'without offering a detailed score.' We have also provided more detailed representation of the CEJST data. The updated paragraph in Methods section is as follows:

“The CEJST offers a binary classification to identify DACs, supported by detailed data attributes that indicate the type and extent of disadvantage across a wide range of indicators. The final DAC classification also includes census tracts that, while not meeting every threshold, are completely surrounded by DACs and have low-income rates at or above the 50th percentile. In contrast, the DOE classification is based on top 20% state percentile rankings derived from a low-income indicator and various categorical thresholds with a focus in energy-related indicators. We chose to use data from the DOE Disadvantaged Community Reporter, as it includes additional energy-related categories that may be relevant to EVCS, which could allow us to better correlate these challenges with user experiences and sentiments regarding EVCS. The substantial overlap between CEJST and DOE DAC data ensures consistency in identifying DACs.”

Page 14, LLM methodology: This is a very novel and interesting application of a LLM. My kudos to the authors.

Thank you for the kudos!

Page 15, lines 507-514: Has this approach with ChatGPT been documented in other studies? If so, this should be cited.

Thank you for your comments. We have added the following reference which demonstrates this approach in research, to the revised manuscript:

Davis, J., Bulck, L. Van, Durieux, B. N. & Lindvall, C. The Temperature Feature of ChatGPT: Modifying Creativity for Clinical Research. *JMIR Hum. Factors* **11**, e53559 (2024).

From the supplemental table, it would appear that GPT 3.5 and GPT 4 had almost the same overall accuracy and precision. Is there a statistically significant difference? This would be helpful to understand since other researchers may have more constrained resources, and if there is no statistically significant difference, GPT 3.5 would be a more parsimonious choice.

You are correct; GPT-3.5 and GPT-4 showed nearly identical overall accuracy, with differences that were not statistically significant. Given that GPT-3.5's sentiment analysis capability is sufficiently accurate and was 20 times cheaper at the time of our research, we recommend using GPT-3.5. In our study, we used GPT-3.5 for batch analysis and tested GPT-4 solely for the validation data sets. To clarify this for readers, we have revised the following discussion in our manuscript:

“We found that, while ChatGPT-4 performed slightly better, the GPT-3.5 model—20 times more cost-effective—also demonstrated strong capabilities for determining the sentiment of EVCS reviews. This highlights the effectiveness of LLMs in sentiment analysis tasks. Given that GPT-4's improvements are focused more on multimodal and visual recognition abilities, we conclude that GPT-3.5 is a more cost-effective choice for sentiment analysis in our study. After validating the model on a subset, we applied GPT-3.5 to the entire dataset (n = 470 142) to identify negative reviews.”

Pages 16-18, Grounded Theory: I can appreciate how detailed the authors were in documenting their process. Since Grounded Theory has different ontological and epistemological underpinnings, it would be helpful for the authors to document which version of grounded theory they are using. It would appear they are working off of Charmaz (2006), but this has not been explicitly stated.

Yes, we used Charmaz version of grounded theory. We have clarified this in the updated Methods section as follows:

“We used Charmaz’s constructivist grounded theory^{6,7} approach, a systematic methodology in social science research that involves constructing theories through interpretive analysis, to analyze the negative sentiment in the reviews (n = 1 081) and categorize them into distinct themes.”

Once this has been established, the authors should mention their QA/QC methodology. A good reference for this is Charmaz and Thornberg (2021).

Thank you for your comments. We have clarified our QA/QC method in a subsection within the grounded theory section, as follows:

“*Quality Assurance and Quality Control (QA/QC)*. To ensure rigor in our grounded theory analysis, we applied QA/QC practices aligned with methodology proposed by Charmaz and Thornberg¹. We maintained a reflective approach, regularly revisiting our methods and findings to ensure they accurately represented user experiences. Initially, two researchers independently coded the reviews, followed by collaborative discussion to

refine subcategories. For example, 'App Crashes' and 'App Problems' were merged into 'App Functionality,' while similar steps helped consolidate other overlapping codes. Throughout the analysis, we followed an iterative process of open, axial, and selective coding to continually reassess and refine our categories. Cross-checking between researchers in each phase helped identify necessary additions, deletions, or merges, ensuring that our categories accurately represented user feedback while avoiding redundancy. As part of our QA/QC process, we validated our findings by comparing our coding categories with three recent studies²⁻⁴ on EVCS reliability. This step aligns with Charmaz and Thornberg's suggestion to engage with existing literature. As a result of this QA/QC process, our results aligned well with all the issues identified in prior studies and revealed additional problem categories users face in public EVCS."

References:

1. Charmaz, K. & Thornberg, R. The pursuit of quality in grounded theory. *Qual. Res. Psychol.* **18**, 305–327 (2021).
2. California Air Resources Board. Electric Vehicle Charging Survey Results. (2023).
3. Karanam, V. C. & Tal, G. How Disruptive are Unreliable Electric Vehicle Chargers? Empirically Evaluating the Impact of Charger Reliability on Driver Experience. (2024) doi:10.2139/SSRN.4752496.
4. Rempel, D., Cullen, C., Bryan, M. M. & Cezar, G. V. Reliability of Open Public Electric Vehicle Direct Current Fast Chargers. *SSRN Electron. J.* (2022) doi:10.2139/SSRN.4077554.

Reviewer #2 (Remarks to the Author):

The paper offers an interesting comparison of DAC and non-DAC areas by analyzing the number of EV charging stations in census tracts and explores user experience using user-generated data. However, while the study utilized valuable datasets, it has significant shortcomings that detract from its contribution to the field and make it unsuitable for publication in its current form.

We thank you for taking the time to review our manuscript. Your comments have been very valuable to us, and we have made significant revisions based on your feedback.

First, while the study aims to investigate "equity of access," neither "equity" nor "access" is adequately defined or thoroughly analyzed. Equity can take on different meanings depending on the perspective considered, yet the paper limits its exploration to a basic comparison of the number of chargers in DAC and non-DAC areas. This approach overlooks the complexity of equity and fails to capture its multi-dimensional nature.

Initially, we used the term 'access' because we believed our approach represented an improvement over existing literature by using real-world data to filter out EVCS locations with access restrictions or payment requirements, as shown in Fig. 1 in the manuscript. In addition, multiple studies we cited in the Introduction used the term 'access' even when the analysis was based mainly on EVCS numbers in a given area. Please note, we keep the word "access" in the Introduction when referring to these previous studies.

However, we agree with you that our methodology primarily analyzes spatial distribution or coverage rather than the true 'access'. We recognize that a comprehensive assessment of 'access' would require additional factors, such as pricing, points of interest, and more. Thus, we now use 'access' only to refer to unrestricted access to EVCS (public). In our revised manuscript, we have changed 'equity of access' to 'equitable coverage', and provided the definition in the updated manuscript as follows:

"We define equitable EVCS coverage as a 3-mile radius around charging stations that are open to the public and have no additional parking fees, based on data from the UGC platform. Studies indicate that most public EV charging events occur within this distance for both DC fast chargers and Level 2 chargers¹. This radius also aligns with the preferred driving distance for EV drivers seeking a charge.^{2,3}"

We also conducted sensitivity analysis of different radius, and included in the method section:

"To assess the robustness of our findings, we conducted a sensitivity analysis by varying the KDE radius to 5 miles and 10 miles. As shown in Table S2, these alternative

radius values did not alter our conclusions, demonstrating that our results are consistent across different spatial scales.”

References:

1. Tal, G. *et al.* Advanced Plug-in Electric Vehicle Travel and Charging Behavior Final Report (CARB Contract 12-319-Funding from CARB and CEC). (2020).
2. Pevec, D. *et al.* A survey-based assessment of how existing and potential electric vehicle owners perceive range anxiety. *J. Clean. Prod.* **276**, 122779 (2020).
3. Philipsen, R., Schmidt, T. & Ziefle, M. Well worth a detour?—Users’ preferences regarding the attributes of fast-charging infrastructure for electromobility. *Adv. Intell. Syst. Comput.* **484**, 937–950 (2017).

Furthermore, the absence of key control variables, such as population density, proximity to points of interest, the number of registered EVs within each tract, and many others, hampers the study to provide a realistic presentation of equity.

In the original manuscript, we controlled population by using per 1 000 population metrics. We agree that incorporating population density, as you suggested, would be valuable. However, other factors, such as the number of registered EVs at the census tract level, are unfortunately not available on a national scale, which is why we did not include them in the analysis

We have now acknowledged these limitations in the Discussion section:

“Limitations. In our analysis, we focused primarily on spatial coverage, while considering whether EVCS locations are publicly accessible and free of additional parking fees. However, we recognize that 'access' to EVCS could be defined more comprehensively, incorporating factors such as charging speed, station capacity, and user behavior (e.g., average charging time and frequency of use). Access to behavioral and charging session data could deepen understanding, though this information is often proprietary and difficult to obtain. Future research might address these limitations by integrating additional accessibility factors, providing a more holistic assessment of EVCS accessibility that includes both infrastructure and operational capacity. Another limitation of this study is the unavailability of nationwide data on the number of registered EVs within each census tract. This data gap restricts our ability to assess EVCS accessibility relative to EV ownership in each community. Future studies could improve precision and apply additional control metrics that adjust for EV ownership density across tracts.”

In my opinion, simply having an equal number of chargers in DAC and non-DAC areas does not necessarily reflect equitable access. Equity involves more than equal distribution and it should account for the specific needs and characteristics of each area. However, if the study considers equal distribution as a measure of equity, it must clearly define what is meant by "equity" and justify why this definition is appropriate in the context of EV charging infrastructure. Without a clear, contextually relevant definition, assuming that an equal number of chargers equates to equitable access is overly simplistic and could be misleading about the true nature of equity in this context. The study reduces "access" to the mere number of EV charging stations without considering essential spatial factors like distance. Since census tracts vary widely in size, simply counting the number of chargers in each tract offers a limited picture. For example, in many previous studies, access is more comprehensively measured by factors like the distance to chargers, which is crucial, especially when considering the significant size variations among census tracts. The findings in this study are likely influenced by population density and the size of these tracts. It is possible that DACs are denser than non-DACs, but without controlling for tract size and density, we cannot know. This is extremely important because dense census tracts tend to be significantly smaller in size, which limits the available public space for EV charging infrastructure (e.g., parking spaces).

Furthermore, dense census tracts often have significantly short distances to neighboring tracts, which could allow residents to rely on chargers outside their tract. This further questions the analysis, as simply counting chargers within a single tract does not provide a reliable measure of access. In short, the current approach does not capture the complexity of equity in access, especially without accounting for spatial (e.g., distance to chargers) and density-related factors.

We sincerely thank you for educating us on this topic, from which we have benefited greatly. We would also like to take this opportunity to share our previous work process. We applied different methodologies while developing the original manuscript, including (1) using only the number of EVCS, (2) calculating the distance to the geometric centroid, and population-weighted centroid for each census tract and determining the average distance to a charging station, and (3) using KDE analysis. We found that each method delivered a similar message: disparities exist between DAC and non-DAC areas. However, we felt that the zonal statistics from KDE and other spatial metrics might not be intuitive for readers unfamiliar with GIS methodologies. As a result, we opted to present only the number of EVCS.

In retrospect, we recognize that using option 1 was too simplistic, and that clearly defining the equity of access is necessary. We have now defined the equity of access in updated manuscript more clearly (see responses above).

We have now included new results that discuss EVCS coverage per capita and MDU considerations (Fig. 2), per square mile (Fig. S2), adjustments for population density (SI Table S1). While the key conclusion remains unchanged, the message is now even stronger: the disparity per 1,000 population increased from 26% to 64% using the KDE method, and further increased to 73% for renters in MDU.

The study mentions that "we present the first national analysis of equity in access to public EV charging ...," which is not accurate. There is already a substantial body of research that has addressed "equity" and "access" to EVCS with great depth (at local and national levels). This is one other major issue of the paper, which is the lack of engagement with the relevant literature, which leaves its contributions unclear and its contextualization within the field weak. I'd suggest that the paper provide more detailed literature on studies that investigated equity and accessibility. This will also help authors come up with a clearer definition of these two terms.

Thank you for your suggestion. We have added five more recent articles related to our work, as follows:

"In Austin, Texas, local analyses¹ found pronounced disparities, with most public chargers concentrated in Non-Hispanic White neighborhoods, underscoring regional patterns of unequal access by race and income. Another study² tracking EVCS distribution from 2010 to 2022 found that, despite a general increase in stations, low-income populations—already underserved by public transportation—continue to lack sufficient EVCS coverage. Similarly, one national study³ found that areas with a higher percentage of racial minorities are less likely to have access to charging stations, while more affluent regions benefit from greater access. Large-scale survey⁴ data have also highlighted the role of socioeconomic factors such as income, age, and housing type in shaping EV adoption and charging behaviors, emphasizing the need for integrated policies addressing both housing and transportation to foster equitable access. A comprehensive review⁵ of national EVCS distribution reinforced these findings, pointing out the persistent gaps in access for lower-income and minority communities and calling for policy frameworks that prioritize equitable infrastructure development to meet 2030 transportation electrification goals. Overall, the gradually expanding body of research on EVCS equity has consistently revealed a striking fact: inequities in EVCS access persist, even as the number of stations continues to increase. This underscores the need for further research on equitable EVCS distribution."

References:

1. Jiao, J., Choi, S. J. & Nguyen, C. Toward an equitable transportation electrification plan: Measuring public electric vehicle charging station access disparities in Austin, Texas. *PLoS One* **19**, e0309302 (2024).

2. Juhasz, L. & Hochmair, H. H. Spatial and Temporal Analysis of Location and Usage of Public Electric Vehicle Charging Infrastructure in the United States. *ISBN Volume 11*, 83–100 (2023).
3. Ermagun, A. & Tian, J. Charging into inequality: A national study of social, economic, and environment correlates of electric vehicle charging stations. *Energy Res. Soc. Sci.* **115**, 103622 (2024).
4. Lee, D. Y., McDermott, M. H., Sovacool, B. K. & Isaac, R. Toward just and equitable mobility: Socioeconomic and perceptual barriers for electric vehicles and charging infrastructure in the United States. *Energy Clim. Chang.* **5**, 100146 (2024).
5. Varghese, A. M., Menon, N. & Ermagun, A. Equitable distribution of electric vehicle charging infrastructure: A systematic review. *Renew. Sustain. Energy Rev.* **206**, 114825 (2024).

We further expanded discussion on Carlton and Sultana (2024) in Discussion section, which was included in the original manuscript. The updates are as follows:

“A recent study¹ has shown that nearly 60-80% of census tracts across the U.S. have no public charging access, except for residents living near highway charging corridors. Many rural areas still lack public charging stations, and the average travel time to EVCS in these regions are longer than urban areas.”

1. Carlton, G. J. & Sultana, S. Electric vehicle charging equity and accessibility: A comprehensive United States policy analysis. *Transp. Res. Part D Transp. Environ.* **129**, 104123 (2024).

We have also softened our tone in the discussion section, as follows:

“We present a national analysis of public EVCS coverage from an equity perspective, using user-generated data that more accurately reflect public accessibility and charger reliability.”

In terms of methodology, the paper would benefit from a better presentation of the data sources and analytical process. The study does not provide sufficient details about the user-generated data platform, which is a critical component of the research.

The data were sourced from PlugShare. We did not include the platform name in the manuscript because another research group, Asensio et al., used PlugShare data in their 2020 publication in Nature Sustainability without naming it either. Although they did

not explicitly mention the platform, the data structure indicates its use. We felt it might be more appropriate not to specify the name to avoid implying an endorsement of PlugShare. At the time of our study, only a commercial license for API usage was available, and an academic license was not despite our attempts. Thus, we utilized publicly available PlugShare data using web crawlers instead of their API, which complies with the platform's guidelines, as their robots.txt file does not restrict such use for non-commercial purposes.

Reference:

Asensio, O. I. *et al.* Real-time data from mobile platforms to evaluate sustainable transportation infrastructure. *Nat. Sustain.* 2020 36 3, 463–471 (2020).

Additionally, the analysis would be more compelling if visualized through maps, and the inclusion of a table summarizing the datasets would benefit the method and data section.

In conclusion, while the paper has potential, especially with its use of user-generated data and LLM, it falls short in several key areas. A more robust analysis, a more comprehensive literature review and definitions of key terms, and better visualization of results are needed to meet the standards of journal publication.

We previously attempted to visualize the data on a map (see below), but many census tracts lack EVCSs, and the number of charging stations varies greatly. As a result, (1) the map appeared cluttered due to extensive grey areas without data, (2) it was challenging to establish an appropriate color range for the legend, and (3) it was impossible to show statistical differences between DACs and non-DACs on this map. Therefore, we decided to display state-level data using a 'pseudo' US map as shown in Fig 2.

Additionally, we have included detailed, summarized state-level data for various metrics in SI Table S1.

Reviewer #2 (Remarks on code availability):

The provided code includes Python scripts for sentiment analysis and categorization. However, the scripts for the other parts of the analysis are not included.

We have now included the Python scripts used to generate Fig. 2 and Figs S1-2, which are more complex visualizations created with Python. Other figures, such as bar and pie charts, are simpler and do not require additional scripts, as they can be easily reproduced using standard plotting functions.

You can access the codes using the same link we provided earlier:

<https://figshare.com/s/209509bbc5067c936ce3>

Reviewer #3 (Remarks to the Author):

This paper explores issues of charging equity in the U.S. It conducts a state-wide analysis and examines the needs of disadvantaged communities, who are noted to predominantly lack adequate access to public EV charging stations. I found the article engaging until I reached the section on the use of a LLM piece. There are several arguments made that appear to have methodological flaws. The reviewer is uncertain whether this can truly be termed 'sentiment analysis,' as it feels more like binary encoding with the LLM assigning labels based on those numbers.

We appreciate the time you've taken to review our paper. However, we respectfully disagree with your interpretation that our method is flawed. Below, we have provided detailed clarifications on our method to offer a clearer understanding of our approach and address your concerns.

First, regarding what is considered as 'sentiment analysis'. Foundational studies^{8,9} frame sentiment analysis as a binary classification problem, simply 0 or 1. Continuous encoding is a method that can be used, but the ultimate goal is often to achieve binary classification. Surveys¹⁰ conducted on sentiment analysis all show that binary encoding are dominant ways of modeling the problem. Continuous modeling of sentiment scores is uncommon, and it is also difficult to calibrate human annotators' judgments using continuous scores.

Second, regarding using 'continuous numbers' as a direct output to assess sentiment analysis from CNN or LLM. The probabilities (often presented as continuous numbers) provided by CNNs (just like other deep learning methods) are not, in fact, trustworthy and this has been recognized to be a fundamental problem in machine learning by multiple papers^{11,12}. Hence, it is not common to directly use probabilities from any architecture relying on deep learning. So, there is no advantage to using CNNs in our study; in either case (LLM or CNN), direct use of probabilities from the model is non-standard. In addition, unlike CNNs, due to the nature of generative AI, while LLMs are effective at generating final sentiment analysis labels, using them to produce continuous numerical outputs to explain their decision-making process can lead to inconsistent results. It's important to note that LLMs cannot be evaluated using the traditional frameworks and ideologies applied to neural network models.

The authors argue for the novelty of using UGC platform and point out potential shortcomings of the NREL database. While the reviewer acknowledges the importance of including UGC, there are likely biases that should be considered, such as gender,

income, age, and language due to the types of users contributing reviews. Typically, in NLP analyses, such as previous studies using Twitter data, irrelevant reviews and outliers are cleansed. This step seems to be missing in the current analysis. Because of the uncertainty surrounding the UGC dataset, it feels like the authors are attempting to combine two papers into one. The first paper should focus on addressing the use of UGC, while the second should concentrate on conducting sentiment analysis using UGC. This creates confusion for readers, as the article mixes both quantitative and qualitative research approaches.

Thank you for raising a valid point regarding potential biases in EVCS reviews due to factors like user demographics (e.g., gender, income, age). Although demographic data are not available in the PlugShare dataset, this data source has been successfully used in previous studies and is recognized for its comprehensive and high-quality user-generated content on EVCS performance and accessibility. For example, a previous study¹³ that used PlugShare data has achieved reliable and representative results, supporting the robustness of our analysis despite the lack of demographic specifics. Additionally, as the largest UGC platform in the U.S. with over 3.5 million users, PlugShare provides access to a broad demographic of EV drivers, enhancing the representativeness of our data. Our study's innovation lies in reaching actual EV drivers—a group often challenging to access through traditional mail-in surveys.

Unlike Twitter data, which are known for poor data quality¹⁴ and can contain up to 53% noise or spam¹⁵, PlugShare data are of higher quality, with fewer than 5% of reviews being incomprehensible. During our data sourcing process, we performed additional cleaning, such as removing empty review comments. This level of quality in PlugShare data minimizes the need for extensive data cleaning processes commonly required for platforms like Twitter, where unreadable messages are more prevalent.

We have added the following details in the methodology section to reduce confusion:

“Unlike social media platforms like Twitter, which may contain bot-generated or incomprehensible comments, the EVCS UGC dataset we selected is of very high quality, as the EVCS community tends to provide meaningful input, as indicated by other research¹³. Empty comments or those containing only single letters or numbers have been filtered out, accounting for less than 5% of the data.”

The authors also argue that the large volume of data justifies the use of an LLM, but the reviewer disagrees with this claim. It is still possible to perform conventional sentiment analysis using NLP techniques. If the use of an LLM is to be justified, it should be compared with conventional methods not just by citing other's work that is trained on different data and research framework.

Thank you for your comment. In the original manuscript lines 455-464, we state that LLM's main advantage (as shown by our results) lies in eliminating the need for feature extraction and model training since it's already pretrained, as compared to traditional NLP model.

Discussion on whether LLMs outperform traditional NLP is beyond the scope of this paper. Our intent is to demonstrate how LLMs reduce the need for extensive preprocessing compared to traditional NLP methods, which could streamline workflows for researchers who may not have the expertise to train NLP models themselves. However, in studies where LLMs were compared to more conventional methods, they were shown to have achieved comparable or superior performance on language-based tasks, including sentiment analysis. For example, one comprehensive study¹⁶ conduct a fairly analysis, testing not only CNNs, but also GNNs, RNNs, and other models. The study makes a compelling case that even earlier generations of language models (such as BERT) were already outperforming these models by significant margins, and newer generations (like Llama 2) were at that level or better. In some other papers (e.g., by Xing¹⁷) it is also alluded to that in specific domains (like financial sentiment analysis) LLMs also achieve excellent performance, without requiring massive training.

Again, we do not claim that LLMs inherently perform better; rather, our intent is to demonstrate that, from an efficiency and economic perspective, LLMs could be a viable alternative.

In conclusion, I am rejecting this article due to the foundational issues with its hypothesis and methods, but I would encourage the authors to further develop their work.

My additional comments are provided below:

Our hypothesis is not that LLMs outperform traditional NLP models; rather, we hypothesize that DACs are underserved by public EVCS.

1) In the introduction section, the authors should provide a clearer definition of DAC and explain how current federal policies like Justice40 address or define them. The problem statement is mainly based on empirical studies in metropolitan regions such as California and NYC. There are other case studies widely available across the U.S., and further descriptions would strengthen the problem statement.

Thank you for your suggestion. We have added more clarification of the DAC and Justice40 program.

“We obtained environmental justice and DAC designation data from the DOE Justice40 initiative via the DOE Disadvantaged Community Reporter (DCR)¹⁸. This dataset includes a set of indicators to identify and characterize DACs. Specifically, the DOE dataset encompasses multiple environmental, economic, and social indicators that reflect the cumulative burdens experienced by different census tracts. Examples of these indicators include air quality, energy burden, and socioeconomic factors¹⁹.

It is important to note that the White House Council on Environmental Quality (CEQ) also provides the Climate and Economic Justice Screening Tool (CEJST)²⁰, a new tool that aims to help federal agencies identify DACs. The CEJST offers a binary classification to identify DACs, supported by detailed data attributes that indicate the type and extent of disadvantage across a wide range of indicators. The final DAC classification also includes census tracts that, while not meeting every threshold, are completely surrounded by DACs and have low-income rates at or above the 50th percentile. In contrast, the DOE classification is based on top 20% state percentile rankings derived from a low-income indicator and various categorical thresholds with a focus in energy-related indicators. We chose to use data from the DOE Disadvantaged Community Reporter, as it includes additional energy-related categories that may be relevant EVCS, which could allow us to better correlate these challenges with user experiences and sentiments regarding EVCS. The substantial overlap between CEJST and DOE DAC data ensures consistency in identifying DACs.”

2) What are the characteristics of users in the UGC platform, including factors like race, age, and income? The reason for this question is that digital platforms are typically more used by younger populations and likely English speakers. While the authors argue that the DOE database has shortcomings, doesn't UGC also carry inherent biases? How do the authors address these issues?

As an online platform, just like Twitter data, user profile data are generally unavailable due to privacy concerns. Similarly, we cannot access such data from PlugShare. However, PlugShare, the largest UGC platform in the U.S. with over 3.5 million users, provides access to a broad demographic of EV drivers, enhancing the representativeness of our data. Mail-in surveys may collect anonymous demographic data, but the response rates are often low, making it even harder to reach actual EV or EVCS users.

Our claim regarding the DOE's database shortcomings was never about inherent biases, as the reviewer may have misinterpreted. In the original manuscript, we stated that EVCS UGC platforms provide 'more detailed site information and user reviews,' which helps us obtain more real-world information, particularly regarding actual accessibility.

Nevertheless, we have reflected the potential user bias in our dataset and added the following in the method section:

“A limitation is that, while this UGC platform effectively reaches actual EV and EVCS users—often difficult to access through traditional mail-in surveys—it may still reflect biases common to online platforms, such as demographic skew related to race, age, and income. Unfortunately, due to privacy restrictions, we cannot access detailed demographic data to adjust for these potential biases. Future work could address this limitation by integrating demographic data through anonymized data-sharing agreements if such data are available.”

3) I liked Figure 2.

Thank you for your positive feedback.

4) This leads to further doubts about whether the authors objectively cleansed outliers. The authors might want to refer to conventional processes for sentiment analysis in NLP. Additionally, EV drivers are mobile, and it's challenging to confirm that a user reviewing a public EV charger in California actually resides there. How did the authors account for this type of bias?

Our LLM analysis focuses on capturing EV user experiences with EVCS, rather than directly assessing accessibility based on user residency or charging location. Consequently, even if a California resident posts a review about an EVCS in Texas, for example, their feedback on charger reliability, user satisfaction, or negative experiences remains valuable for understanding user experience across regions.

5) The reviewer believes that the review categories used to prompt the LLM in Table 2 should be supported not only by data but also by a comprehensive literature review.

We appreciate the reviewer's suggestion regarding the review categories used to prompt the LLM. In fact, we conducted a QA/QC process when developing these problem categories, employing grounded theory to ensure a rigorous approach. We have now explicitly included a QA/QC section in the methodology, detailing how we compared our categories with existing literature to validate our approach:

“Quality Assurance and Quality Control (QA/QC). To ensure rigor in our grounded theory analysis, we applied Quality Assurance and Quality Control practices aligned with

methodology proposed by Charmaz and Thornberg²¹. We maintained a reflective approach, regularly revisiting our methods and findings to ensure they accurately represented user experiences. Initially, two researchers independently coded the reviews, followed by collaborative discussion to refine subcategories. For example, 'App Crashes' and 'App Problems' were merged into 'App Functionality,' while similar steps helped consolidate other overlapping codes. Throughout the analysis, we followed an iterative process of open, axial, and selective coding to continually reassess and refine our categories. Cross-checking between researchers in each phase helped identify necessary additions, deletions, or merges, ensuring that our categories accurately represented user feedback while avoiding redundancy. As part of our QA/QC process, we validated our findings by comparing our coding categories with three recent studies²²⁻²⁴ on EVCS reliability. This step aligns with Charmaz and Thornberg's suggestion to engage with existing literature. As a result of this QA/QC process, our results aligned well with all the issues identified in prior studies and revealed additional problem categories users face in public EVCS."

6) Instead of explicitly assigning binary labels, the authors could benefit from assigning continuous sentiment scores to measure relativity on a continuous scale.

As we mentioned in the first paragraph, assigning continuous sentiment scores (1) is uncommon, (2) complicates and reduces consistency in the model validation process, and (3) is not well-suited to LLMs, which are generative by nature and not optimized for producing continuous sentiment scores to explain the decision-making process. Unlike traditional NLP models that are often designed to output specific and discrete sentiment scores, LLMs generate responses based on contextual understanding, which can make precise scoring more variable and challenging to standardize.

7) The authors mention that two reviewers were assigned to the code. Since this involves qualitative aspects, did the authors obtain IRB approval? Shouldn't the positionality of the reviewers also be acknowledged? Is it typical to have only two reviewers?

A qualitative aspect does not necessarily require IRB approval; IRB oversight is specifically needed when research involves human subjects. In our research, all data are secondary, with no interviews or surveys administered; therefore, IRB approval is not required.

We have trained our annotators on common ground so they won't have personal positions as stated in line 490-492 in the original manuscript. As shown in previous

studies, it is typical to use two annotators to test inter-rater reliability in this type of analysis^{25,26}.

8) The authors selected a sample size of 3,000, but the reviewer disagrees that this is a robust sampling approach. Instead, sampling should involve multiple iterations, and the results should be generalized based on the compilation of those iterations.

As explained in lines 472-474 in the original manuscript, our annotated sample size is robust within the field^{27,28} of sentiment analysis validation. See table below (next page) adopted from Gohil et al. In the machine learning literature, including sentiment analysis, it is standard to sample and sequester a 'test set' of withheld data only once, and use it to obtain an unbiased, statistically high-powered estimate of performance. Our test set was fully withheld from the system doing the sentiment analysis prediction and reserved exclusively for the evaluation.

Table 2. Sentiment tools based on type of tool: KNN: k-nearest-neighbors; N/A: not applicable; NB: Naïve Bayes; SVM; support vector machines.

Author	Tool	Annotators	Kappa	Manually annotated sample
Cole-Lewis et al [29]	Produced for study: machine learning classifiers based on 5 categories (NB, KNN, and SVM)	6	.64	250
Desai et al [31]	Produced for study: rule based using AFINN (Named after the author, Finn Arup Neilsen)	N/A	N/A	N/A
Daniulaityte et al [30]	Produced for study: logistic regression, NB, SVM	2	.68	3000
Myslin et al [34]	Produced for study: machine learning (NB, KNN, SVM)	2	>.7	1000
Sofean and Smith [36]	Produced for study: 5-fold validation using support vector machines (SVM's) model using Waikato Environment for Knowledge Analysis toolkit toolkit	N/A	N/A	500
Tighe et al [37]	Produced for study: rule based using AFINN	N/A	N/A	N/A
Bhattacharya et al [27]	Open source: SentiStrength	3	N/A	N/A
Hawkins et al [33]	Open source: machine learning classifier using Python library TextBlob	2+Amazon Mechanical Turk	>.79	2216
Ramagopalan et al [26]	Open source: TwitterR R package + Jeffrey Breen's sentiment analysis code	N/A	N/A	N/A
Black et al [28]	Commercial: radian6	N/A	N/A	N/A
Greaves et al [32]	Commercial: TheySay	N/A	N/A	250
Nwosu et al [35]	Open source: TopsyPro	N/A	N/A	N/A

9) Sentiment analysis results for the entire state should be presented in maps. Figure 3a currently illustrates the status of the U.S. rather than providing a breakdown across individual states.

Thank you for the suggestion. However, since sentiment analysis results are closely related to urban and rural classifications, as shown in Figure 3 of the original manuscript, displaying results by individual states would largely reflect this urban-rural divide. We believe the current figure provides clear and sufficient information, and an additional state-level figure would not add additional insights.

10) In Table S2, the authors attempt to justify the use of GPT-3.5 or GPT-4 over conventional NLP methods and cite Asenio et al. (2020). This is incorrect. It is not appropriate to claim one model is superior to another unless they are trained on the same datasets. For example, in the article, GPT-4 achieves 87.1% accuracy, but if another study using CNN achieves 87.2% accuracy, would that mean CNN is superior to GPT-4 in sentiment analysis? The authors should conduct a separate NLP analysis and compare it with GPT-3.5 and GPT-4.

Thank you for your diligence in reviewing our methods. In fact, the dataset used by Asensio et al. is also sourced from PlugShare, the same dataset we used in our analysis. Given this alignment, we believe the comparison is appropriate.

Again, our goal is not to show that LLMs perform better than CNNs; rather, we aim to demonstrate that LLMs offer a valid and feasible approach for EVCS review sentiment analysis.

11) This article is submitted to Nature Communications. Given the aim and scope of the journal, how can the findings be generalized to a global audience?

We believe our findings hold relevance for a global audience, even though they are based on a dataset from the United States. *Nature Communications* regularly publishes regional studies that inform decision-making in a broader context. Our paper addresses a universally relevant topic: to facilitate widespread EV adoption ('the egg'), accessible EVCS infrastructure ('the chicken') is essential. Additionally, we demonstrate the potential of LLMs in qualitative data analysis, showcasing AI's utility for societal good, which aligns with the interests of the journal's broad scientific readership. We believe *Nature Communications* is an ideal platform to disseminate our work.

References:

7. Foley, G. & Timonen, V. Using Grounded Theory Method to Capture and Analyze Health Care Experiences. *Health Serv. Res.* **50**, 1195–1210 (2015).
8. Pang, B., Lee, L. & Vaithyanathan, S. Thumbs up?: sentiment classification using machine learning techniques. 79–86 (2002) doi:10.3115/1118693.1118704.
9. Turney, P. D. Thumbs up or thumbs down? semantic orientation applied to unsupervised classification of reviews. *Proc. Annu. Meet. Assoc. Comput. Linguist.* **2002-July**, 417–424 (2002).
10. Birjali, M., Kasri, M. & Beni-Hssane, A. A comprehensive survey on sentiment analysis: Approaches, challenges and trends. *Knowledge-Based Syst.* **226**, (2021).
11. Abdar, M. *et al.* A review of uncertainty quantification in deep learning: Techniques, applications and challenges. *Inf. Fusion* **76**, 243–297 (2021).
12. Gawlikowski, J. *et al.* A survey of uncertainty in deep neural networks. *Artif. Intell. Rev.* **56**, 1513–1589 (2023).
13. Asensio, O. I. *et al.* Real-time data from mobile platforms to evaluate sustainable transportation infrastructure. *Nat. Sustain.* **2020 36 3**, 463–471 (2020).
14. Murshed, B. A. H., Mallappa, S., Ghaleb, O. A. M. & Al-ariki, H. D. E. Efficient Twitter Data Cleansing Model for Data Analysis of the Pandemic Tweets. *Stud. Syst. Decis. Control* **348**, 93–114 (2021).
15. Tsou, M.-H., Zhang, H. & Jung, C.-T. Identifying Data Noises, User Biases, and System Errors in Geo-tagged Twitter Messages (Tweets). *arXiv.org* (2017).
16. Sharma, N. A., Ali, A. B. M. S. & Kabir, M. A. A review of sentiment analysis: tasks, applications, and deep learning techniques. *Int. J. Data Sci. Anal.* (2024) doi:10.1007/s41060-024-00594-x.
17. Xing, F. Designing Heterogeneous LLM Agents for Financial Sentiment Analysis. *ACM Trans. Manag. Inf. Syst.* (2024) doi:10.1145/3688399.
18. U.S. Department of Energy. Energy Justice Mapping Tool - Disadvantaged Communities Reporter. <https://energyjustice.egs.anl.gov/> (2024).
19. Office of Energy Justice and Equity. Justice40 Initiative | Department of Energy. <https://www.energy.gov/justice/justice40-initiative> (2023).
20. White House Council on Environmental Quality. Climate & Economic Justice Screening Tool. <https://screeningtool.geoplatform.gov/en/#3/33.47/-97.5> (2022).
21. Charmaz, K. & Thornberg, R. The pursuit of quality in grounded theory. *Qual. Res. Psychol.* **18**, 305–327 (2021).
22. California Air Resources Board. Electric Vehicle Charging Survey Results. (2023).
23. Rempel, D., Cullen, C., Bryan, M. M. & Cezar, G. V. Reliability of Open Public

- Electric Vehicle Direct Current Fast Chargers. *SSRN Electron. J.* (2022) doi:10.2139/SSRN.4077554.
24. Karanam, V. C. & Tal, G. How Disruptive are Unreliable Electric Vehicle Chargers? Empirically Evaluating the Impact of Charger Reliability on Driver Experience. (2024) doi:10.2139/SSRN.4752496.
 25. Wiebe, J., Wilson, T. & Cardie, C. Annotating expressions of opinions and emotions in language. *Lang. Resour. Eval.* **39**, 165–210 (2005).
 26. Pang, B. & Lee, L. A Sentimental Education: Sentiment Analysis Using Subjectivity Summarization Based on Minimum Cuts. *Proc. 42nd Annu. Meet. Assoc. Comput. Linguist.* (2005) doi:https://doi.org/10.3115/1218955.1218990.
 27. van Atteveldt, W., van der Velden, M. A. C. G. & Boukes, M. The Validity of Sentiment Analysis: Comparing Manual Annotation, Crowd-Coding, Dictionary Approaches, and Machine Learning Algorithms. *Commun. Methods Meas.* **15**, 121–140 (2021).
 28. Gohil, S., Vuik, S. & Darzi, A. Sentiment Analysis of Health Care Tweets: Review of the Methods Used. doi:10.2196/publichealth.5789.

Reviewer #3 (Remarks on code availability):

N/A

All comments from the reviewers have been addressed below point-by-point.

Responses are in blue color and manuscript changes are also included in this file and highlighted in yellow in the revised manuscript.

Reviewers' comments:

Reviewer #1 (Remarks to the Author):

I appreciate the time and effort the authors invested in addressing my comments and concerns during the review process. Their detailed revisions, including the incorporation of new references, expansion on the methodology, and careful attention to ensuring the reliability of the data and its presentation, are commendable. While I cannot speak for the other reviewers, it seems that the authors also gave due consideration to critiques raised by others, including Reviewer 3's detailed comments about the LLM. Given these thorough efforts and improvements, I am of the opinion that this article is now suitable for publication in Nature Communications.

We want to sincerely thank you again for your time and effort in reviewing our revisions and for your thoughtful comments throughout the process. We deeply appreciate your recognition and endorsement of our paper, which means a great deal to us.

Reviewer #3 (Remarks to the Author):

I appreciate the authors' efforts and the time they dedicated to improving the overall quality of their manuscript. While there are some points on which I disagree, the manuscript appears suitable for publication in its current form.

I acknowledge that leveraging LLMs and AI agents is a trending topic in empirical research; however, I encourage the authors to carefully reflect on its theoretical contributions in their future work. Overall, great work.

Thank you for taking the time to review our paper again. We value the opportunity to exchange differing perspectives and ideas, and we appreciate your recognition of the quality of our manuscript.

Reviewer #3 (Remarks on code availability):

I have reviewed them.

Thank you for reviewing our codes.

Reviewer #4 (Remarks to the Author):

I was not one of the original referees, but the editor has asked me to review the new submission along with the reviewer reports to arbitrate between them. This is easier than I expected, but (a) the study is a first-rate contribution to the fields of energy and transport equity and (b) the authors have done a good-faith job responding to all of the earlier comments, even those from the critical referees. For these reasons, I believe the paper is of publishable quality and I recommend that the critical review be overruled. Thank you for taking the time to review our manuscript, as well as the comments and responses from the previous round of peer review. We sincerely appreciate your recognition of our work and your support for its publication.

In reviewing the manuscript, I did have a series of Minor Revisions to suggest which would improve it before it is published.

In the Abstract, the authors highlight that "disadvantaged communities (DACs) have 64% fewer public EVCSs per capita than non-DACs" and that "users in DACs and 12 urban areas experience significantly more reliability issues compared to those in non-13 DACs and rural areas." This begs the question for me, however, which is what % of EVCS's actually experience reliability issues (especially negative user experiences) overall vs. in DACs. I say this because it's a very novel finding and also one I have experienced myself - I do not own a car but have tried renting an EV twice and experienced many EVCS reliability issues where I live in New England.

Thank you for your question. The average user experience rankings for DAC and non-DAC areas are presented in Fig. 3a. To address your query, we calculated the percentage of EVCS experiencing reliability issues based on several Wilson score thresholds. Wilson score is effective for handling a small number of reviews across many stations, and then ranking the scores for each station. The results are presented in the table below, which shows that, across all thresholds, DACs have a higher number or percentage of EVCS considered unreliable.

Wilson score threshold	Unreliable EVCS% in DAC	Unreliable EVCS% overall
0.1	40.7%	36.9%
0.2	57.2%	53.1%
0.3	75.2%	73.3%

However, we don't believe it is appropriate to set an arbitrary threshold on the Wilson score to define an unreliable EVCS. Therefore, we decided not to include specific numbers in the revised manuscript. However, our main conclusion remains the same.

Second, the article's two core themes are equity and reliability, but these are (obviously) concerned with vehicle use and charging infrastructure. Other equity and justice considerations are manifold and include other dimensions (procedural justice, dependence on automobility, recognitional justice, even issues of mineral extraction)

well beyond the vehicle or charging station. It would be worth mentioning this in the Limitations section, that the study takes a narrower focus on equity, and that future work could explore (with large datasets such as the study's) these other justice concerns. Especially the impacts from energy transition metal mining, supply chain and labor issues in EV manufacturing, and pollution flows after end of life. To support these claims, see:

Henderson, Jason. "EVs are not the answer: a mobility justice critique of electric vehicle transitions." *Annals of the American Association of Geographers* 110, no. 6 (2020): 1993-2010.

Skeete, Jean-Paul, Peter Wells, Xue Dong, Oliver Heidrich, and Gavin Harper. "Beyond the EVent horizon: Battery waste, recycling, and sustainability in the United Kingdom electric vehicle transition." *Energy Research & Social Science* 69 (2020): 101581.

Sovacool, Benjamin K., Johannes Kester, Lance Noel, and Gerardo Zarazua de Rubens. "Energy injustice and Nordic electric mobility: Inequality, elitism, and externalities in the electrification of vehicle-to-grid (V2G) transport." *Ecological economics* 157 (2019): 205-217.

Sovacool, B.K., Hook, A., Martiskainen, M. and Baker, L., 2019. The whole systems energy injustice of four European low-carbon transitions. *Global Environmental Change*, 58, p.101958.

Svobodova, Kamila, John R. Owen, Deanna Kemp, Vítězslav Moudrý, et al. "Decarbonization, population disruption and resource inventories in the global energy transition." *Nature communications* 13, no. 1 (2022): 7674.

Thank you for your suggestions; we agree that EVs are not a solution to all challenges. We appreciate your suggestion to address this in the limitations section as part of future research directions.

We have now incorporated the following discussion in the limitations section of the manuscript, referencing the five recommended citations:

“Finally, from a broader equity and social justice perspective, it is important to acknowledge that EVs alone cannot address all transportation-related challenges, despite their significant potential for decarbonizing the transportation sector in combating climate change. Research⁷³ has highlighted the need for comprehensive solutions, including systematic changes to reduce car dependency and develop more equitable mobility options. Additionally, concerns⁷⁴ have been raised regarding the environmental and social impacts of battery production, global waste inequality throughout the lifecycle stages, and broader justice implications of the clean energy transition—including how renewable energy development⁷⁵ affects local communities and the impacts on traditional carbon-intensive industries and their workforces⁷⁶.

Returning to this paper's focus on EV charging infrastructure, future research and policy directions must emphasize procedural justice to achieve genuine equity in infrastructure deployment. Study⁷⁷ indicates that without a comprehensive justice framework encompassing distributive and recognition justice components, the transition to EVs may perpetuate or even exacerbate existing inequalities. This is especially critical as residents of DACs are often excluded from the policy-making process during the clean energy transition^{69,78}. A holistic approach^{79,80} is essential, incorporating elements of recognition and procedural justice.”

References:

73. Henderson, Jason. "EVs are not the answer: a mobility justice critique of electric vehicle transitions." *Annals of the American Association of Geographers* 110, no. 6 (2020): 1993-2010.
74. Skeete, Jean-Paul, Peter Wells, Xue Dong, Oliver Heidrich, and Gavin Harper. "Beyond the Event horizon: Battery waste, recycling, and sustainability in the United Kingdom electric vehicle transition." *Energy Research & Social Science* 69 (2020): 101581.
75. Sovacool, B.K., Hook, A., Martiskainen, M. and Baker, L., 2019. The whole systems energy injustice of four European low-carbon transitions. *Global Environmental Change*, 58, p.101958.
76. Svobodova, Kamila, John R. Owen, Deanna Kemp, Vítězslav Moudrý, et al. "Decarbonization, population disruption and resource inventories in the global energy transition." *Nature communications* 13, no. 1 (2022): 7674.
77. Sovacool, Benjamin K., Johannes Kester, Lance Noel, and Gerardo Zarazua de Rubens. "Energy injustice and Nordic electric mobility: Inequality, elitism, and externalities in the electrification of vehicle-to-grid (V2G) transport." *Ecological economics* 157 (2019): 205-217.

Third, the study is very up to date on its citations but it is missing one more very recent one from DY Lee and his team:

Lee, D. Y., Wilson, A., McDermott, M. H.,... & Ward, J. (2025). Does electric mobility display racial or income disparities? Quantifying inequality in the distribution of electric vehicle adoption and charging infrastructure in the United States. *Applied Energy*, 378, 124795.

Thank you for the recommendation, we have now discussed and included this 2025 publication as follows:

“A recent study²⁹ highlights that income and racial/ethnic inequalities in charging infrastructure distribution are significantly greater than those for gas stations, with disparities varying widely across states and urbanized areas.”

In addition, we cited another recent 2025 EVCS related publication as follows:

“Even with corridor charging infrastructure programs, rural areas still remain underserved⁵⁷.”

References:

29. Lee, D. Y., Wilson, A., McDermott, M. H.,... & Ward, J. (2025). Does electric mobility display racial or income disparities? Quantifying inequality in the distribution of electric vehicle adoption and charging infrastructure in the United States. *Applied Energy*, 378, 124795.

57. Hanig, Lily, Ledna, Catherine, Nock, Destenie, Harper, Corey D., Yip, Arthur, Wood, Eric, & Spurlock, C. Anna. Finding gaps in the national electric vehicle charging station coverage of the United States. *Nat. Commun.* 2025 161 **16**, 1–13 (2025).

Fourth, in Table 1 “Summary of current issues for electric vehicle charging stations (EVCSs) 284 based on negative user reviews,” it would be good to add a final column indicating how frequent these issues come up from the data (either as a total N= or as a %).

Thank you for your suggestion. We have already presented the different problem categories as percentages in Fig. 3b. However, we do not have the detailed counts for specific issues. This is because classifications with more labels can reduce both inter-rater consistency and LLM performance. Thus, we provided the issue types as context and only asked the annotators and the LLM to classify the reviews into the 5 broader problem categories instead of the 17 issue types.

Otherwise, a solid study that I look forward to citing in my own work.

Thank you again for your recognition of our work!